# Temperature and volumetric effects on structural and dielectric properties of hybrid perovskites

Andrzej Nowok [1,2] ✉, Szymon Sobczak [3], Kinga Roszak [3], Anna Z. Szeremeta[4], Mirosław Mączka [5], Andrzej Katrusiak [3] ✉, Sebastian Pawlus[4], Filip Formalik[6,7], Antonio José Barros dos Santos [8], Waldeci Paraguassu[8] & Adam Sieradzki [2] ✉

Three-dimensional organic-inorganic perovskites are rapidly evolving materials with diverse applications. This study focuses on their two representatives - acetamidinium manganese(II) formate (AceMn) and formamidinium manganese(II) formate (FMDMn) – subjected to varying temperature and pressure. We show that AceMn undergoes atypical pressure-induced structural transformations at room temperature, increasing the symmetry from ambient-pressure $P2_1/n$ phase II to the high-pressure $Pbca$ phase III. In turn, FMDMn in its $C2/c$ phase II displays temperature- and pressure-induced ordering of cage cations that proceeds without changing the phase symmetry or energy barriers. The FMD$^+$ cations do not order under constant volume across the pressure-temperature plane, despite similar pressure and temperature evolution of the unit-cell parameters. Temperature and pressure affect the cage cations differently, which is particularly pronounced in their relaxation dynamics seen by dielectric spectroscopy. Their motion require a rearrangement of the metal-formate framework, resulting in the energy and volumetric barriers defined by temperature-independent activation energy and activation volume parameters. As this process is phonon-assisted, the relaxation time is strongly temperature-dependent. Consequently, relaxation times do not scale with unit-cell volume nor H-bond lengths in formates, offering the possibility of tuning their electronic properties by external stimuli (like temperature or pressure) even without any structural changes.

Three-dimensional hybrid organic-inorganic perovskites (3D HOIPs) have emerged as a promising and fast-developing class of materials with a vast array of application possibilities across diverse scientific and technological domains. Sensing, catalysis, gas storage, magnetism, luminescence, optoelectronics, or photovoltaics exemplify some of their diverse applications[1–7]. The key to their success lies in a tuneable framework composed of metal cations coordinated by bridging ligands (organic or inorganic), with cavities occupied by rationally chosen cation[4].

Of particular interest are the HCOO$^-$ anions, used as linkers in hybrid formates. Despite their small size, formate ions can accommodate relatively large cations, such as *bis*(3-ammoniumpropyl) ammonium, N,N'-dimethylethylenediammonium and 1,4-butane-diammonium, without transforming the 3D cage architecture into

lower-dimensional systems[8–10]. Within these structures, the cage cations may exhibit additional dynamic disorder, with their reorientation or hopping between symmetry-equivalent positions triggered by external stimuli, giving rise to orientational polarization and dielectric relaxation[11]. Furthermore, the HCOO⁻ ions are able to pass a canted antiferromagnetic interaction, leading to a weak-ferromagnetic order (canted antiferromagnetic order)[4]. Finally, formate anions can bind metal ions in three different coordination modes: *syn-syn*, *syn-anti*, or *anti-anti*[4,11]. Consequently, formate-bridged hybrid materials exhibit a diverse range of structural, electric, and magnetic properties, encompassing order-disorder phases with ferroic or multiferroic features, and ferroelasticity[12–14].

Despite extensive research efforts dedicated to the study of hybrid formates, the realm of dielectric relaxations in these materials still encompasses numerous unexplained phenomena. For instance, recent high-pressure dielectric studies have revealed a small sensitivity of relaxation times (and relaxation peak position) to compression in several 3D hybrid formates[12–14]. In the case of 1,4-butanediammonium zinc formate (DABZn), the isothermal compression up to 1.7 GPa at 302 K exerts a similar effect on relaxation times as a mere 17 K decrease in temperature (to 285 K)[14]. These findings are rather unexpected, considering the mechanism underpinning the relaxation phenomenon in these materials and the significant softness of their ionic crystal structures, evidenced by relatively small Young's moduli and considerable compressibility[15–18]. This softness is also evident in a moderate pressure regime (below 1 GPa), where most of the pressure-induced phase transition in such materials was observed[19,20]. It should be mentioned in this regard, that the precise role of the dynamics of cage cations in stimuli-triggered structural transformations continues to be an area of intensive studies[13].

To further explore the relaxation phenomena in hybrid formates, we have reinvestigated the structures and dielectric response of their two model derivatives, acetamidinium manganese(II) formate ([NH₂–C(CH₃)–NH₂][Mn(HCOO)₃], further abbreviated as AceMn) and formamidinium manganese(II) formate ([NH₂–CH–NH₂][Mn(HCOO)₃], named as FMDMn). In both compounds, manganese Mn²⁺ cations and formate HCOO⁻ anions form a 3D framework counterbalanced by formamidinium (FMD⁺, [NH₂–CH–NH₂]⁺) or acetamidinium (Ace⁺, [NH₂–C(CH₃)–NH₂]⁺) respectively for FMDMn and AceMn[21,22].

Pressure and temperature are thermodynamic parameters that are frequently used to understand the structure-property relationships in hybrid compounds[20,23,24]. The compression can also induce various phenomena in these materials, e.g., amorphization, tricritical points, polarization enhancement, and metallization[25–31]. In this article, we demonstrate several unconventional pressure-induced effects, such as a phase transition increasing the symmetry, or an order-disorder transformation with no crystal symmetry change nor increased energy barrier. We also derive general relationships that link dielectric features with structural changes in these prototypical HOIPs. In this

regard, we redefine the activation volume parameter and highlight its critical role in regulating relaxation dynamics under pressure. The research is based on the combination of varied-temperature and high-pressure X-ray diffraction (XRD), Raman, and dielectric spectroscopy studies, and supported by quantum density functional theory (DFT) calculations.

## Results

The model compounds FMDMn and AceMn were synthesized as crystalline materials. Detailed crystallographic data under varied-temperature and high-pressure conditions are presented in Tables 1, 2, and Supplementary Tables 1–8. The X-ray diffraction studies show that the FMDMn above 335 K and at ambient pressure is trigonal (space group $R\bar{3}c$), whereas a high-temperature phase I of AceMn stable above room temperature (~300 K) is orthorhombic (space group *Imma*), which corroborates the previous reports (c.f. Tables 1, 2 and Supplementary Table 2)[21,22].

At the ambient conditions ($T = 296$ K, $p = 0.1$ MPa), the FMDMn crystals are monoclinic of space group $C2/c$ (phase II, see Table 1), as reported previously[21]. Under these conditions, the cage cations FMD⁺ are disordered with the site occupation factor (SOF) equal to 0.85 and 0.15. Isothermal compression of phase II reveals no anomalies in the unit-cell dimensions up to 3.63 GPa (Fig. 1a). The softest of the unit-cell parameters is $a$, which, together with increasing angle $\beta$, is responsible for the diagonal contraction of the perovskite cages (see Fig. 1a, b). This mechanism involving the tilts of MnO₆ octahedra reduces the void space in the unit cell and gradually eliminates the disorder in phase II, forcing the FMD⁺ cations to stack into one energetically privileged position. This phenomenon is manifested in the pressure evolution of SOF parameters: the larger SOF₁ gradually increases and the smaller SOF₂ decreases on compression, approaching 1 and 0, respectively (Fig. 1c). Ultimately, the SOF parameter remains constant above 1 GPa, while any fluctuations are within the experimental errors. (Fig. 1c). This progressive change does not affect the crystal symmetry as FMD⁺ cations assume the position with the strongest H-bonds to the formate ligands. This provides additional support for the framework, righting the crystal direction [010] along which the H-bonds are aligned, and resulting in almost no compressibility along this direction up to 3.63 GPa (the parameter $b$ is only reduced by about 1%). The linear compressibility along [010] direction calculated as $\beta_y = -1/b\ \partial b/\partial p$ for compression range from 1.09 GPa to 1.95 GPa, is close to zero, with $\beta_y = 0.39(3)$ TPa⁻¹ (see Supplementary Tables 9 and 10). Interestingly, the strain tensor at 3.63 GPa reveals the negative linear compressibility (NLC) of −4.2(5) TPa⁻¹ in the direction close to [102] (see Supplementary Table 9). This effect can be explained by the wine-rack mechanism[32], with the strongest compression along the shortest diagonal of the rhomboidal cages and the elongation along the longest diagonal[33–35]. Above 4 GPa the

## Table 1 | Selected unit-cell parameters of FMDMn in its phases I and II under various pressure–temperature conditions

|  |  | Phase I[21] | Phase II | Phase II | Phase II |
|---|---|---|---|---|---|
| **Space group** |  | $R\bar{3}c$ | $C2/c$ | $C2/c$ | $C2/c$ |
| **Temperature** |  | 355 K | 296 K | 110 K | 296 K |
| **Pressure** |  | 0.1 MPa | 0.1 MPa | 0.1 MPa | 1.09 GPa |
| **Unit-cell parameters (Å)** | $a$ | 13.8035 (9) | 13.8051 (9) | 13.4375 (7) | 12.92 (2) |
|  | $b$ | 8.8776 (2) | 8.6947 (3) | 8.6838 (4) | 8.6313 (6) |
|  | $c$ | 8.8776 (2) | 8.4685 (5) | 8.4075 (4) | 8.237 (6) |
| **$\beta$ angle (°)** |  | - | 119.715 (8) | 118.15 (1) | 116.14 (14) |
| **Volume (Å³)** |  | 1337.2 (1) | 882.82 (11) | 865.0 (1) | 824.6 (18) |
| **Z/Z′** |  | 6/0.1667 | 4/0.5 | 4/0.5 | 4/0.5 |

## Table 2 | Selected unit cell parameters of AceMn in its phases I, II, and III under various pressure–temperature conditions

|  |  | Phase I | Phase II | Phase III |
|---|---|---|---|---|
| **Space group** |  | *Imma* | $P2_1/n$ | *Pbca* |
| **Temperature** |  | 300 K | 296 K | 296 K |
| **Pressure** |  | 0.1 MPa | 0.1 MPa | 1.34 GPa |
| **Unit-cell parameters (Å)** | $a$ | 8.7433 (3) | 8.7440 (4) | 11.888 (8) |
|  | $b$ | 12.0592 (4) | 12.0563 (4) | 11.674 (3) |
|  | $c$ | 9.0164 (3) | 9.0175 (4) Å | 12.819 (3) |
| **$\beta$ angle (°)** |  | 90 | 90.099 (5) | 90 |
| **Volume (Å³)** |  | 950.66 (6) | 950.63 (7) | 1779.1 (13) |
| **Z/Z′** |  | 1/0.0625 | 1/0.25 | 2/0.25 |

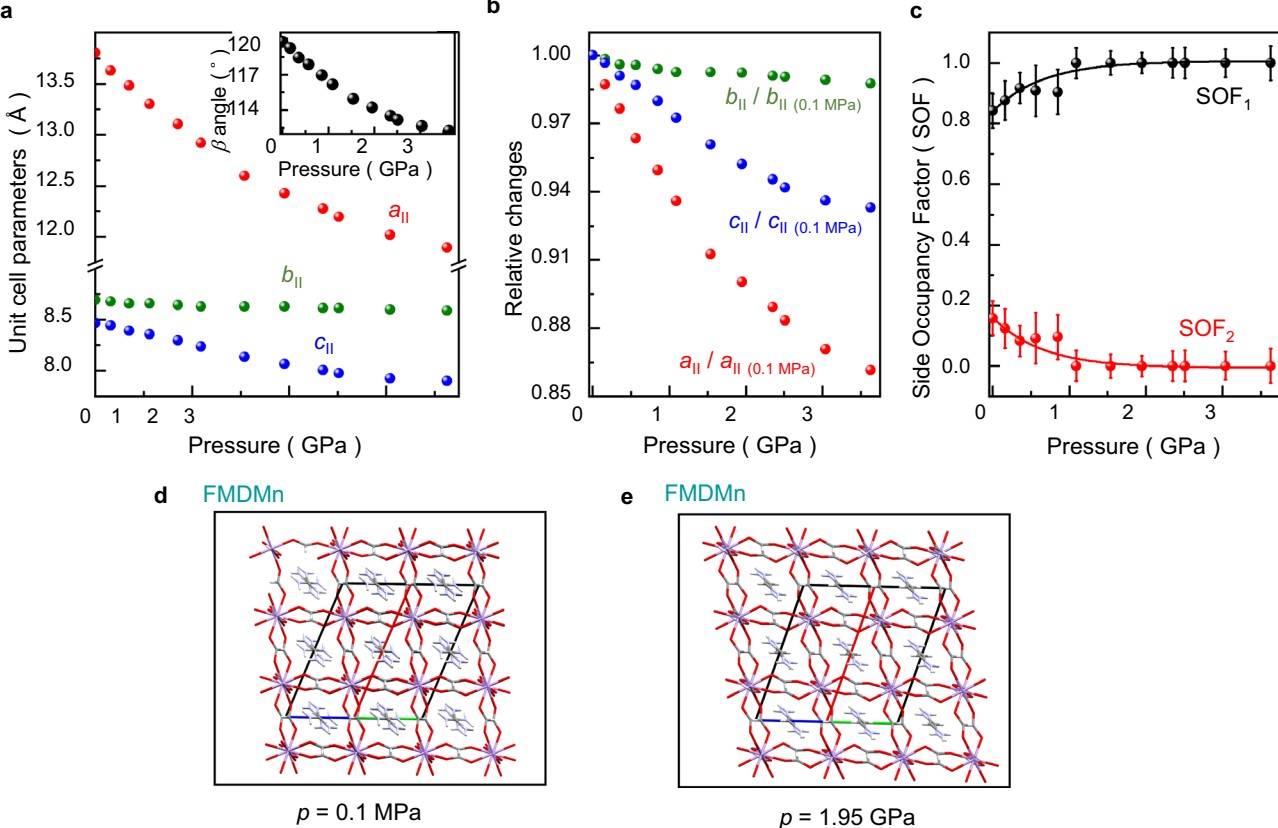

**Fig. 1 | Crystal structure of FMDMn under ambient and high pressure.**
**a** Variation of unit cell parameters $a_{II}$, $b_{II}$, $c_{II}$ during room-temperature isothermal compression of FMDMn. Upper inset shows pressure-induced changes in $\beta$ angle.
**b** Relative changes in unit cell parameters $a_{II}$, $b_{II}$, $c_{II}$ observed during the compression of FMDMn at room temperature. The changes were calculated relative to the ambient-pressure values of these parameters ($a_{II\,(0.1MPa)}$, $b_{II\,0.1MPa}$, $c_{II\,0.1MPa}$).
**c** Changes in SOF parameter of FMDMn as a function of pressure. Error bars represent the standard deviation. The structure of FMDMn at room temperature at pressures (**d**) 0.1 MPa and (**e**) 1.95 MPa plotted along $b$ axis. Mn, O, C, N, and H atoms are marked in purple, red, gray, blue, and white, respectively. Source data are provided as a Source Data file.

intensity of the diffraction spots decreases significantly. We carried out high-pressure Raman measurements at room temperature to explain this observation.

Raman spectra collected during room-temperature compression of FMDMn are presented in Supplementary Fig. 1 and the corresponding pressure dependence of Raman wavenumbers together with assignment based on previous studies of formamidinium-based formates is shown in Supplementary Fig. 2. The values of wavenumber intercepts at zero pressure ($\omega_0$) and pressure coefficients ($\alpha = d\omega/dp$) obtained from fitting of the experimental data with a linear function $\omega(p) = \omega_0 + \alpha p$ are listed in Supplementary Table 11. This table also summarizes the assignment of these bands according to previous studies on formamidinium-based formates[21,36]. As presented in Supplementary Fig. 1, the Raman spectra remain qualitatively the same up to about 0.6 GPa. When the pressure reaches 0.9 GPa, the intensity of the two strongest lattice bands near 160 and 107 cm⁻¹ starts to decrease at the expense of the bands at 175 and 125 cm⁻¹ (values at 0.9 GPa, see Supplementary Fig. 1a). This behavior continues beyond 0.9 GPa. Strong intensity increase is also observed for the 81, 1377, and 1386 cm⁻¹ bands, which become clearly seen above 0.9 GPa (Supplementary Fig. 1a, b). Since the X-ray diffraction evidenced no phase transition up to 3.63 GPa, it was noted that the pressure at which Raman spectra change (~0.9 GPa) correlates with complete ordering of FMD⁺ cations near about 1 GPa (Supplementary Fig. 1c). The analysis of the Raman data show that nearly all modes in the 0–3.5 GPa range exhibit a positive pressure dependence and that the largest pressure coefficients $\alpha$ are observed for the lattice modes, especially those involving Mn²⁺ translations (Supplementary Table 11). This

behavior indicates that the compression involves tilts of MnO₆ octahedra, in agreement with the X-ray diffraction. The negative pressure dependence is observed for the C–N stretching modes near 1150 cm⁻¹ only (see Supplementary Table 11). This behavior can be attributed to the elongation of C–N bond and the decrease of the N–C–N bond angles on compression.

When the pressure reaches 4.0 GPa, some bands are significantly broadened (see for instance the $\nu_3$(HCOO⁻) and $\nu_3$(HCOO⁻) bands at 797 and 1368 cm⁻¹) while others split e.g. $\delta$(CN) + $\tau$(NH₂) band near 595 cm⁻¹ (Supplementary Figs. 1 and 2b). Supplementary Fig. 2b also shows that the C–N stretching modes change slopes of wavenumbers vs pressure from strongly negative below 4 GPa to positive or weakly negative above 4 GPa. All these changes involve both HCOO⁻ and FMD⁺ units, indicating that FMDMn experiences a structural phase transition near 4 GPa to a lower symmetry phase associated with distortion of the framework and increased FMD-framework interactions. Consequently, the weakening of the diffraction peaks near 4 GPa can be associated with this phase transition and attributed to the deterioration of the crystal quality induced by the structural transformation.

On further compression, changes in relative intensities of bands and the appearance of new bands indicate the onset of the third pressure-induced phase transition associated with weak structural changes (see Supplementary Figs. 1 and 2, as well as Supplementary Table 11). Supplementary Fig. 3 shows that during the decompression run, the initial phase is retained. It shows that the pressure-induced phase transitions are reversible and supports the conclusion that FMDMn does not amorphize in the studied pressure range.

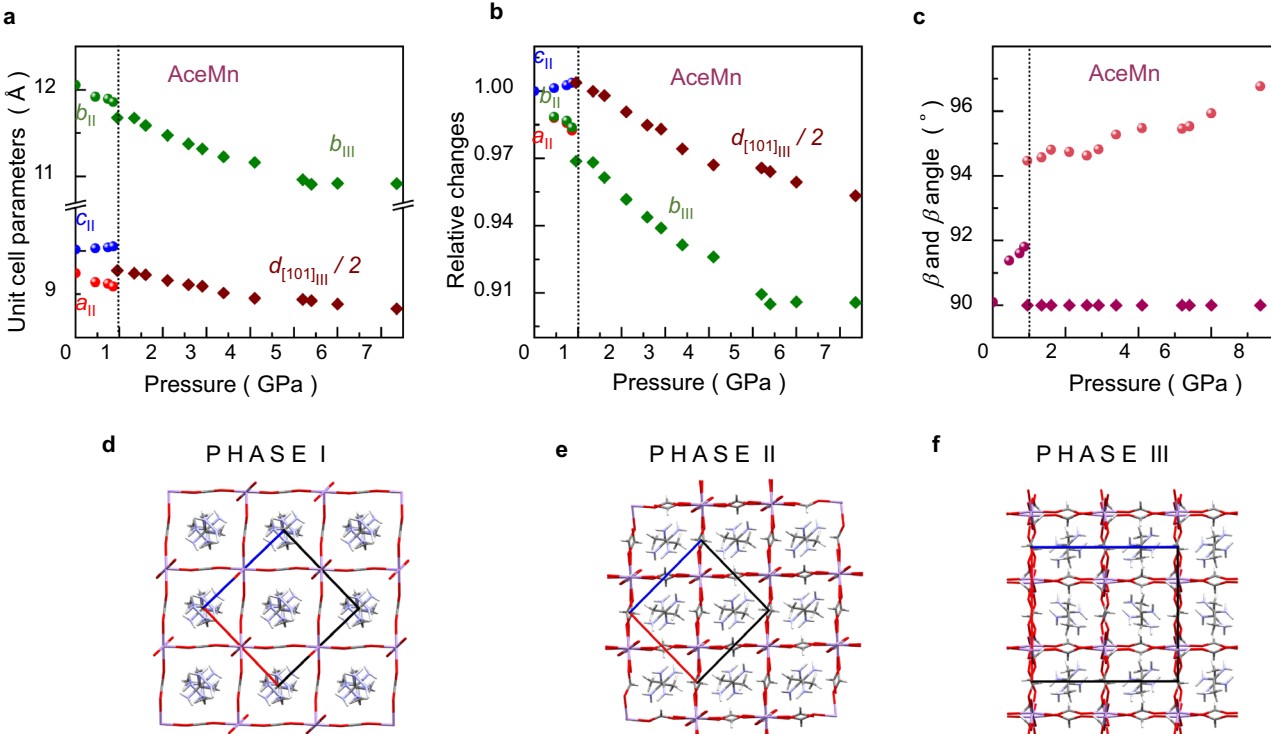

**Fig. 2 | Crystal structure of AceMn under ambient and high pressure. a** Variation of unit-cell parameters $a$, $b$, $c$, $d_{[101]}$ during isothermal compression of AceMn at 298 K. Lower indices II and III in these parameters refer to the phase number. **b** Relative changes (referred to as the STP structure) of the unit-cell parameters $a$, $b$, $c$, $d_{[101]}$ observed while compressing the AceMn crystal at room temperature.

**c** Pressure dependence of the monoclinic angle $\beta_{II}$ in phase, the angle $\beta_{II}$ in the lattice of phase III, and the orthorhombic angle $\beta_{III}$ of phase III. The structure of AceMn in phases I (**d**), II (**e**), and III (**f**) plotted along $b$ axis. Mn, O, C, N, and H atoms are marked in purple, red, gray, blue, and white, respectively. Source data are provided as a Source Data file.

Compared to FMDMn, the behavior of AceMn is much more complex despite the structural similarity of both compounds at ambient pressure (see Fig. 2a–f). At the ambient temperature and pressure, the AceMn crystals are monoclinic of space group $P2_1/n$ (phase II, see Table 2 and Supplementary Table 2), in agreement with previous studies[22]. The monotonic compressibility of phase II involves the NLC with $\beta_c = -9.7(3)$ TPa$^{-1}$ at 0.86 GPa (Fig. 2a, b and Supplementary Table 12), originating from the analogous mechanism as that observed in FMDMn. This NLC effect remains a relatively rare phenomenon among three-dimensional HOIPs. However, there are some notable examples of materials that exhibit NLC, such as chiral $NH_4Zn(HCOO)_3$ ($\beta_x = -1.8(8)$ TPa$^{-1}$)[37] and $[FA]Mn(H_2POO)_3$ ($\beta_x = -7.8(6)$ TPa$^{-1}$)[38]. In comparison, the NLC of AceMn is larger than that of $NH_4Zn(HCOO)_3$. The significance of large NLC phenomena in HOIPs arouses wide interest as it allows to correlate the elongation of the crystal at specific pressure with various other effects, including changes in optoelectronic properties[39]. Such a correlation is important for developing multimodal sensors that can find applications in a wide range of fields, including materials science, nanotechnology, and other technologies[39–43].

Phase II remains stable up to roughly 1.0 GPa when an abrupt volume drop marks a pressure-induced transition of clearly first-order type to phase III of orthorhombic space group $Pbca$ (Fig. 2c). At the critical pressure, the molecular volume (i.e. unit-cell volume per structural unit) is abruptly reduced by about 16 Å$^3$, indicating that the transition involves the collapse of voids in the crystal structure. The presence and first-order character of the phase transition near 1.0 GPa is consistent with previous high-pressure Raman studies, which revealed sudden changes in the spectra between 0.7 and 1.4 GPa[44]. The structural transformation doubles

the unit cell of phase III, as described by the matrix:

$$\begin{pmatrix} a_{III} \\ b_{III} \\ c_{III} \end{pmatrix} = \begin{pmatrix} 1 & 0 & -1 \\ 0 & 1 & 0 \\ 1 & 0 & 1 \end{pmatrix} \begin{pmatrix} a_{II} \\ b_{II} \\ c_{II} \end{pmatrix} \tag{1}$$

Hence, the unit-cell contents double, too (the $Z$ number increases to 2 in phase III, see Supplementary Fig. 4). It is an atypical feature of the transition from AceMn phase II to the high-pressure phase III that its symmetry increases. In most cases, the symmetry of molecular crystals is reduced in order to increase the number of degrees of freedom in the structure, so it efficiently reduces the strain associated with the compression. There are only few exceptions where pressure-induced transitions increase symmetry[45,46]. In AceMn, the monoclinic angle $\beta_{II}$ of phase II, as expected, increases its opening from 90.099(5)° at 0.1 MPa to 91.81(5)° at 0.86 GPa (Fig. 2c). The transition to phase III at 0.9 GPa abruptly increases angle $\beta_{II}$ to 94.47(6)°, but at the same time the unit-cell parameters $a_{II}$ and $c_{II}$ become equal in length (Fig. 2a). It indicates the transition to the orthorhombic system, where directions $[a_{II}]$ and $[c_{II}]$ become the diagonals $d[101]_{III}$ and $d[10\text{-}1]_{III}$ of the doubled unit cell of phase III (Fig. 2e, f). Transformation of AceMn from the low-density monoclinic $P2_1/n$ phase II to the high-density phase III of orthorhombic $Pbca$ space group also changes material compressibility as evidenced by 2$^{nd}$ Birch–Murnaghan coefficient changing from 26.12(1) to 20.0(1) GPa$^{-1}$, respectively (compare Supplementary Tables 12–15).

The transformation between phases II and III preserves the ambient-pressure H-bonding pattern as well as the alignment of the Ace$^+$ cation, consistently with previous high-pressure Raman data,

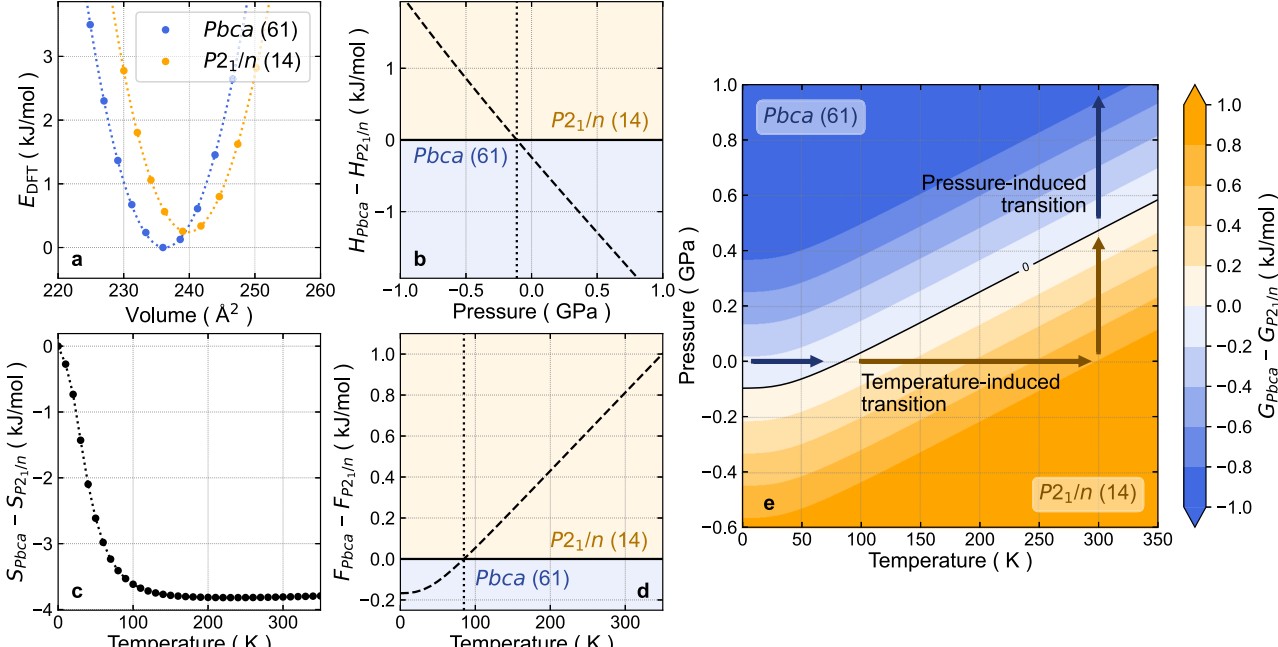

**Fig. 3 | Relative stability of *Pbca* and *P2₁/n* phases in AceMn. a** Energy vs. volume relation for phases II (orange curve and points) and III (blue curve and points) of AceMn. **b** Relative stability of structures with $P2_1/n$ and $Pbca$ symmetries in AceMn under various pressure conditions without considering temperature effects (enthalpy). Here, $H_{Pbca}$ and $H_{P2_1/n}$ denote enthalpies of phases III and II of AceMn, respectively. Areas where $H_{Pbca} > H_{P2_1/n}$ and $H_{Pbca} < H_{P2_1/n}$ are marked in orange and blue, respectively. The dashed line defines the phase transition conditions and the dotted vertical line marks the transition pressure at 0 K. **c** Difference in entropy between $P2_1/n$ and $Pbca$ symmetries versus temperature, highlighting the significance of entropy effects to the phase transition. Here, $S_{Pbca}$ and $S_{P2_1/n}$ denote

entropy of phases III and II of AceMn, respectively. **d** Temperature dependence of Helmholtz free energy, indicating that the monoclinic phase becomes stable at 85 K. Symbols $F_{Pbca}$ and $F_{P2_1/n}$ denote Helmholtz free energy for phases III and II of AceMn, respectively. Areas where $F_{Pbca} > F_{P2_1/n}$ and $F_{Pbca} < F_{P2_1/n}$ are marked in orange and blue, respectively. The dashed line defines the phase transition conditions and the dotted vertical line marks the transition temperature at 0 GPa. **e** Gibbs free energy map for phases II and III in AceMn, revealing their mutual stability under various pressure–temperature conditions. Symbols $G_{Pbca}$ and $G_{P2_1/n}$ denote Gibbs free energy for phases III and II of AceMn, respectively. The black solid line defines the phase transition conditions.

showing that the phase transition weakly affects $\tau NH_2$ and $\rho NH_2$ modes near $544 + 575$ and $1140\,cm^{-1}$, respectively[44]. These relatively strong N−H···O bonds support the structure of phase III up to about 5.0 GPa, when its amorphization occurs. This observation is consistent with previously reported Raman studies, which revealed a sluggish transition in AceMn starting near 5.3 GPa and completed at 8.5 GPa[44]. The formation of the new phase agrees with the progressing increase of the interaction strength between Ace⁺ and the perovskite framework, as evidenced by strong shifts of the $\nu_s CCN$, $\nu_{as} CCN$ and $\nu NCN$ Raman modes to higher wavenumbers (from 906, 1178, and $1524\,cm^{-1}$ at 5.3 GPa to 933, 1195 and $1555\,cm^{-1}$ at 8.5 GPa)[44]. Above 5.0 GPa, the compression breaks the H-bonds and the framework gradually collapses about the cations. Thus the mechanism of the strain compensation in phase III involves the rotations and deformation of formate linkers translating onto the $MnO_6$ octahedra rotations (described in Glazer notation as $a^0a^0c^-$).

The previously reported Raman spectra indicated that the internal vibrations of the $HCOO^-$ ligands and the lattice modes on releasing pressure from 10 GPa to ambient pressure became similar to those observed for the initial ambient-pressure phase[44]. To investigate it and resolve the nature of this hidden phase, we have quickly compressed a AceMn single crystal to the non-hydrostatic region of Daphne 7575 oil (which freezes above 4.5 GPa)[47]. This quick compression preserves the high crystallinity of the sample due to the effect of over-compressing phase III. Then, by gently heating the sample to 320 K, we recovered the hydrostatic conditions in the diamond anvil cell (DAC), as the heating of Daphne oil is a well-established method for restoring its hydrostatic properties[47]. This procedure allowed us to characterize the crystal structure at 7.35 GPa, and then after decompression at 6.00 and 5.20 GPa. The single-crystal diffraction data collected at these

pressures showed that phase III is metastable above 5 GPa. The metastability of phase III was confirmed by a gradual decrease in the intensity of reflections progressing over a period of 2 weeks, due to a slow amorphization process. We have investigated the residual phase III during the amorphization process and the XRD data collected on the sample equilibrated for one week at 5.30 GPa resulted in a unit-cell volume larger than the initial measurement before. The increased unit-cell dimension of residual phase III by ca. 1.3% suggests a mosaic topology of amorphous higher-density regions.

To further elucidate the mechanism behind the structural pressure-induced phase transition in AceMn, we conducted periodic DFT calculations. In this case, we considered only two phases with symmetry $P2_1/n$ (phase II) and $Pbca$ (phase III) to explain their stability at various temperature-pressure conditions. The geometry optimization revealed that the orthorhombic phase II is more stable than phase III when excluding temperature and pressure effects. Specifically, the calculated DFT energy of phase III is approximately 0.25 kJ/mol per formula unit lower than that obtained for phase II. As illustrated in Fig. 3a, phase III remains stable until a volume of about 239 Å³, beyond which phase II becomes stable. A common tangent line constructed from the volume-energy relationship in Fig. 3a demonstrates the pressure at which the transition from phase III to II occurs:

$$p(V) = -\frac{\partial E(V)}{\partial V} \qquad (2)$$

This line is equivalent to the pressure at which enthalpies of both phases ($H = E + pV$) are the same. The calculated transition pressure is approximately −0.11 GPa (see Fig. 3b), which contradicts experimental

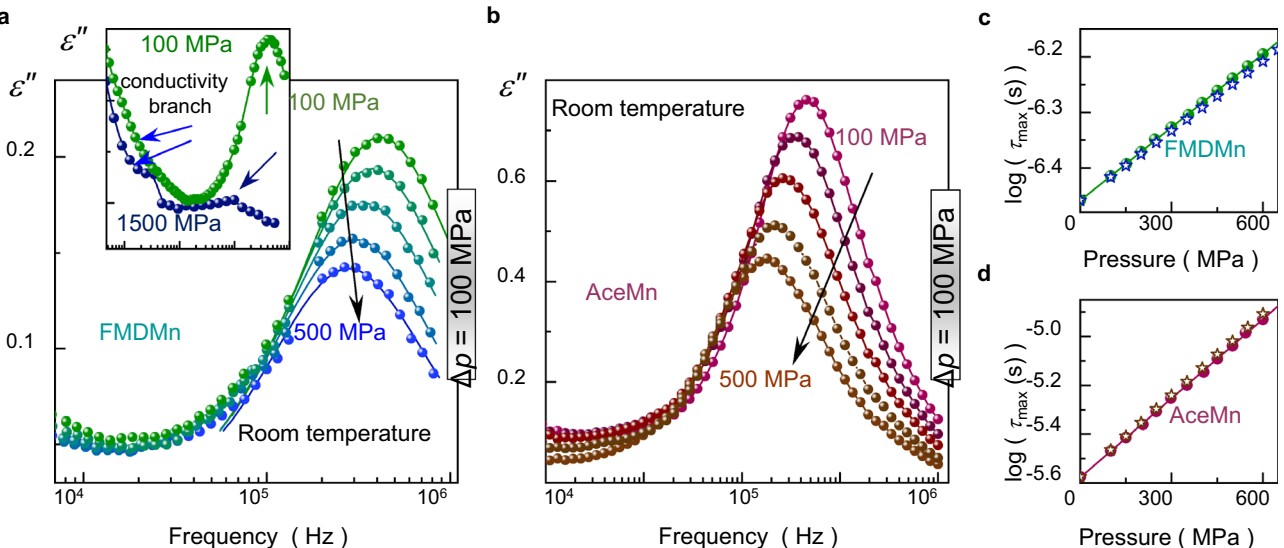

**Fig. 4 | High-pressure dielectric data for FMDMn and AceMn. a** Representative dielectric loss $\varepsilon''(f)$ spectra collected between 100 and 500 MPa at 298 K for FMDMn. Experimental data are represented by colored dots, and Havriliak–Negami fitting functions are shown as colored lines. The upper inset illustrates the relaxation peak vanishing during isothermal compression of FMDMn from 100 MPa (green points) to 1500 MPa (navy dots) at 298 K. **b** Representative $\varepsilon''(f)$ spectra collected during compression at 298 K for AceMn, each coded with different color for clarity. **c** Comparison of experimentally determined (green circles) and calculated (open blue stars) pressure dependence of relaxation times ($\log\tau_{max}$) for FMDMn at 298 K. The green solid line represents the fit of experimental data with Eq. (4). **d** Experimentally determined (red circles) and calculated (open brown stars) pressure dependence of $\log\tau_{max}$ for AceMn at 298 K. The solid line represents the fit of experimental data with Eq. (4). Source data are provided as a Source Data file.

observations and suggests that both temperature and pressure significantly influence the transition.

To incorporate temperature effects, phonon calculations were performed to assess the vibrational partition function and analyze the Helmholtz free energy of both phases. As illustrated in Fig. 3c, d, phase II becomes more stable around 85 K due to increased entropy arising from the mobility of the counterion in the cavity. The low-frequency regime of the phonon density of states, particularly below 2 THz associated with collective rotation-like vibrations of the ion, is more pronounced in the monoclinic phase (see Supplementary Fig. 7). These lower frequencies require less energy to activate the corresponding vibrations of the Ace⁺. The pore size analysis performed on the ion-free frameworks further confirms this observation and shows that the monoclinic phase features larger cages (by approx. 0.2 Å), providing more freedom for countercation movement at increased temperatures (see Supplementary Fig. 8).

Finally, the combined influence of temperature and pressure on the free energy surface is presented in Fig. 3e as a Gibbs free energy map. This landscape should not be treated as a full phase diagram for AceMn as it considers only two phases (II and III) and, thus, can only be used to explain their relative stabilities. Discrepancies in exact values of the phase transition pressure are attributed to the harmonic approximation used in phonon calculations and the potential overestimation of dispersion forces in the Grimme dispersion correction noted in the literature[48].

In terms of dielectric response, a common feature of FMDMn and AceMn is a single relaxation process, which (according to previous studies) originates from a field-induced motion of FMD⁺ and Ace⁺ cations inside the manganese-formate framework, respectively[12,13,21,22]. This relaxation manifests itself as a bell-shaped peak in the imaginary part of the complex dielectric permittivity, $\varepsilon''(f)$, which amplitude decreases steadily during isothermal compression at 298 K for both compounds (see Fig. 4a, b and Supplementary Fig. 5). Eventually, only residual trace of the relaxation process is detectable for FMDMn when 1 GPa is exceeded (see upper inset in Fig. 4a). This trend agrees with the pressure-induced stepwise reduction of disordered FMD⁺ cations

around 1 GPa, as confirmed by high-pressure diffraction studies. The room-temperature ordering of the cage cations under pressure occurs when the unit-cell volume reaches 824.6(18) Å³. In turn, a fully ordered crystal structure of FMDMn at ambient pressure is observed at 110 K when the unit-cell volume is 865.0(7) Å³[21]. Different volume requirements for this process under various $p–T$ conditions indicate that thermal energy is a critical parameter that controls the cage organic cations ordering in HOIPs. Moreover, a direct correlation between crystal structure and dielectric response of FMDMn proves previous hypotheses that gradual ordering of the organic cage cations due to contraction of the cage-like framework constitutes one of the possible origins of pressure-induced diminishing in relaxation peak amplitude in this 3D HOIPs[12,13].

Apart from the amplitude-related effect, progressive shifting of the relaxation process towards lower frequencies occurs for FMDMn and AceMn during their isothermal compression. It means that the related relaxation times ($\tau_{max}$) extend when the pressure increases since $\tau_{max} = \frac{1}{2\pi f_{max}}$[49]. In this formula, $f_{max}$ is the frequency of the loss peak maximum. In order to obtain the pressure dependence of $\tau_{max}$ at 298 K, we parametrize the dielectric spectra of both compounds with the Havriliak–Negami fit function[50] (see Supplementary Note 1 for more details) and calculate the $\tau_{max}$ values based on the fitting parameters $\tau_{HN}$, $\alpha$, $\beta$ according to the formula[49]:

$$\tau_{max} = \tau_{HN}\left[\sin\left(\frac{\alpha\pi}{2\beta+2}\right)\right]^{-\frac{1}{\alpha}}\left[\sin\left(\frac{\alpha\beta\pi}{2\beta+2}\right)\right]^{\frac{1}{\alpha}}. \quad (3)$$

As presented in in Fig. 4c, d, the room-temperature $\tau_{max}(p)$ dependence exhibits an Arrhenius-like character for both FMDMn and AceMn, that can be parametrized by the equation:

$$\tau_{max}(p) = \tau_{max}(0.1\text{MPa})\exp\left(\frac{pV_a}{RT}\right), \quad (4)$$

where $\tau_{0.1\text{MPa}}$ is the ambient-pressure relaxation time, $R$ is the gas constant, and $V_a$ is the activation volume[51]. The determined $V_a$

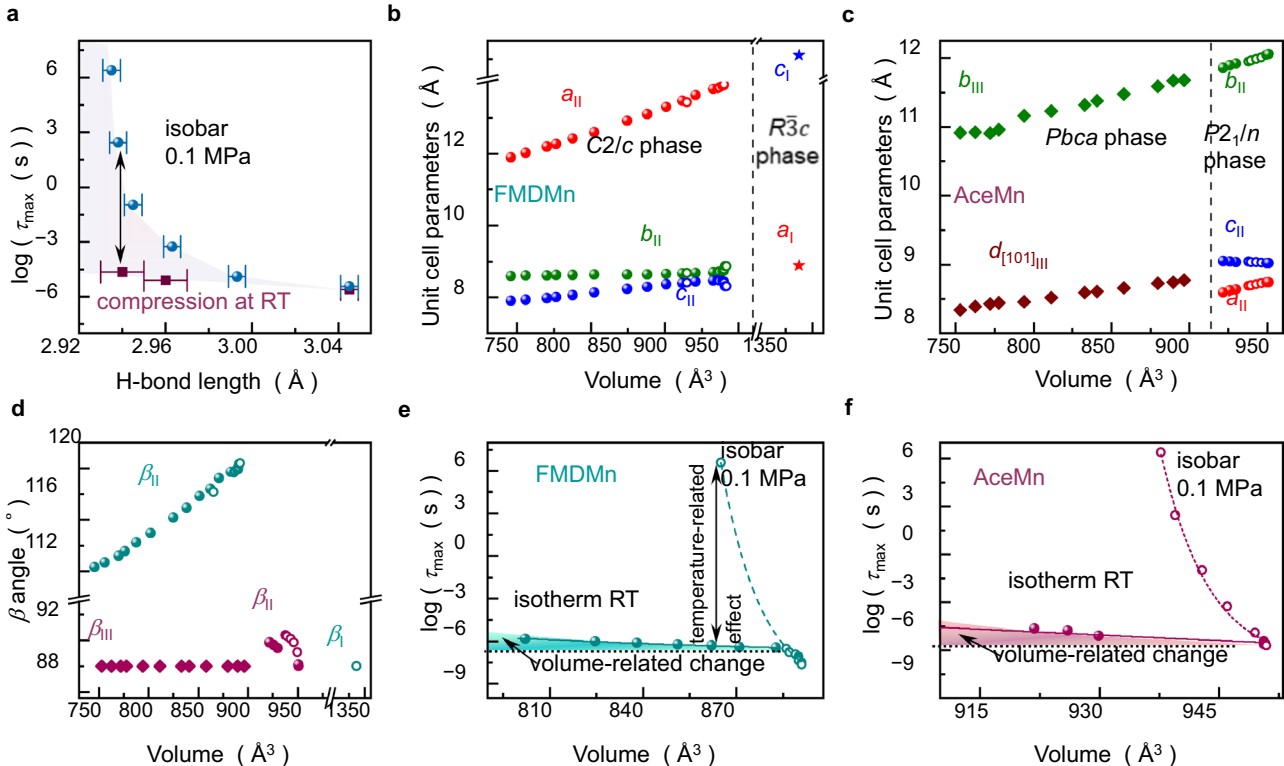

**Fig. 5 | Pressure- and temperature-related effects in FMDMn and AceMn.**
**a** Changes in relaxation times ($\log\tau_{max}$) observed in AceMn during room-temperature compression (red squares) and ambient-pressure cooling (blue dots) plotted versus the length of the O6···H-N2 hydrogen bond. Error bars represent standard deviations. The shaded area defines the difference between these two thermodynamic paths. **b**, **c** Variation of unit-cell parameters *a*, *b*, *c*, $d_{[101]}$ during isothermal compression at 298 K (closed symbols) and isobaric changes in temperature under 0.1 MPa (open symbols) for FMDMn and AceMn. Lower indices I, II, and III in these parameters refer to the phase number. **d** Changes in the $\beta$ angle for FMDMn (blue symbols) and AceMn (red symbols) observed during isothermal compression at 298 K (closed points) and isobaric changes in temperature under 0.1 MPa (open symbols). Lower indices I, II, and III in the $\beta$ parameter refer to the phase number. **e** Dependence of relaxation times on unit-cell volume of FMDMn observed under isobaric conditions of 0.1 MPa (open blue symbols and dotted line) and isothermal high-pressure compression at 298 K (closed blue symbols and solid line). The shaded area indicates volume-related changes. **f** Dependence of relaxation times on unit-cell volume of AceMn observed under isobaric conditions of 0.1 MPa (open symbols and dotted line) and isothermal high-pressure compression at 298 K (closed symbols and solid line). The shaded area defines volume-related changes. Source data are provided as a Source Data file.

parameter takes the value of $2.6 \pm 0.1$ and $6.6 \pm 0.1\,\mathrm{cm^3\,mol^{-1}}$ for FMDMn and AceMn, respectively, being in agreement with previous studies on these compounds[12,13]. What is more, both determined $\tau_{max}(p)$ dependences can be well described by the previously determined equations of state:

$$\tau_{max}(T,p) = \tau_0 \exp\left(\frac{pV_a + E_a}{RT}\right), \qquad (5)$$

(see open points in Fig. 4c, d)[12,13,52]. Here, $\tau_0$ is a pre-exponential factor, $E_a$ denotes the activation energy, and parameters $R$, $V_a$ do not change the meaning relative to Eq. (4). Consequently, we use further this formalism to derive general rules that govern ambient- and high-pressure relaxation dynamics in these hybrid perovskites.

## Discussion

Hydrogen bonds are widely recognized as key forces governing the relaxation dynamics in HOIPs. Therefore, we begin our discussion by clarifying the relationship between their length and relaxation times of the reorientating molecular cations.

As exemplified by AceMn, $\log\tau_{max}$ gradually increases with the shortening (strengthening) of hydrogen bonds (cf. Fig. 5a). This trend is observed during both isobaric cooling and isothermal compression of this material (see blue and red points, respectively). However, there is a significant disparity between these two thermodynamic paths. The compression up to 860 MPa of the hydrogen

bond O6···H-N2 shortens it from 3.05 Å to 2.94 Å at room temperature and increases in $\tau_{max}$ from ~2.46 μs to ~22.7 μs. In contrast, the same change of the hydrogen bonds during isobaric cooling increases $\tau_{max}$ up to approximately 100 s. It is evident that relaxation times in AceMn do not scale with the length of hydrogen bonds, indicating that their strength alone cannot be considered the sole determinant of the relaxation dynamics of Ace⁺ cations. A similar observation can also be made for FMDMn.

Furthermore, the pressure–temperature evolution of $\tau_{max}$ does not correlate molecular volume of FMDMn and AceMn. The X-ray diffraction studies reveal that both FMDMn and AceMn possess relatively soft ionic structures. In AceMn, lowering the temperature from 296 K to 130 K results in a reduction of the unit-cell volume from approximately 950.63 Å³ to roughly 937.73 Å³ while compression at room temperature up to 350 MPa reduces the unit-cell volume to 929.82 Å³. FMDMn shows a similar behavior, with a reduction of the unit-cell volume from approximately 882.20 Å³ to 865.03 Å³ when the temperature is lowered from 300 K to 100 K, while a comparable decrease to 861.91 Å³ is observed during room-temperature compression to 350 MPa. Notably, in both compounds, the variations of the unit-cell parameters $a_0$, $b_0$, and $c_0$ exhibit the same pattern during isobaric cooling (represented by open points) and isothermal compression (represented by closed points). This feature is evident when plotting the $a_0$, $b_0$, and $c_0$, parameters as a function of the unit-cell volume (see Fig. 5b, c). Only minor differences between these thermodynamic paths are noted in the behavior of the $\beta$ angle for AceMn

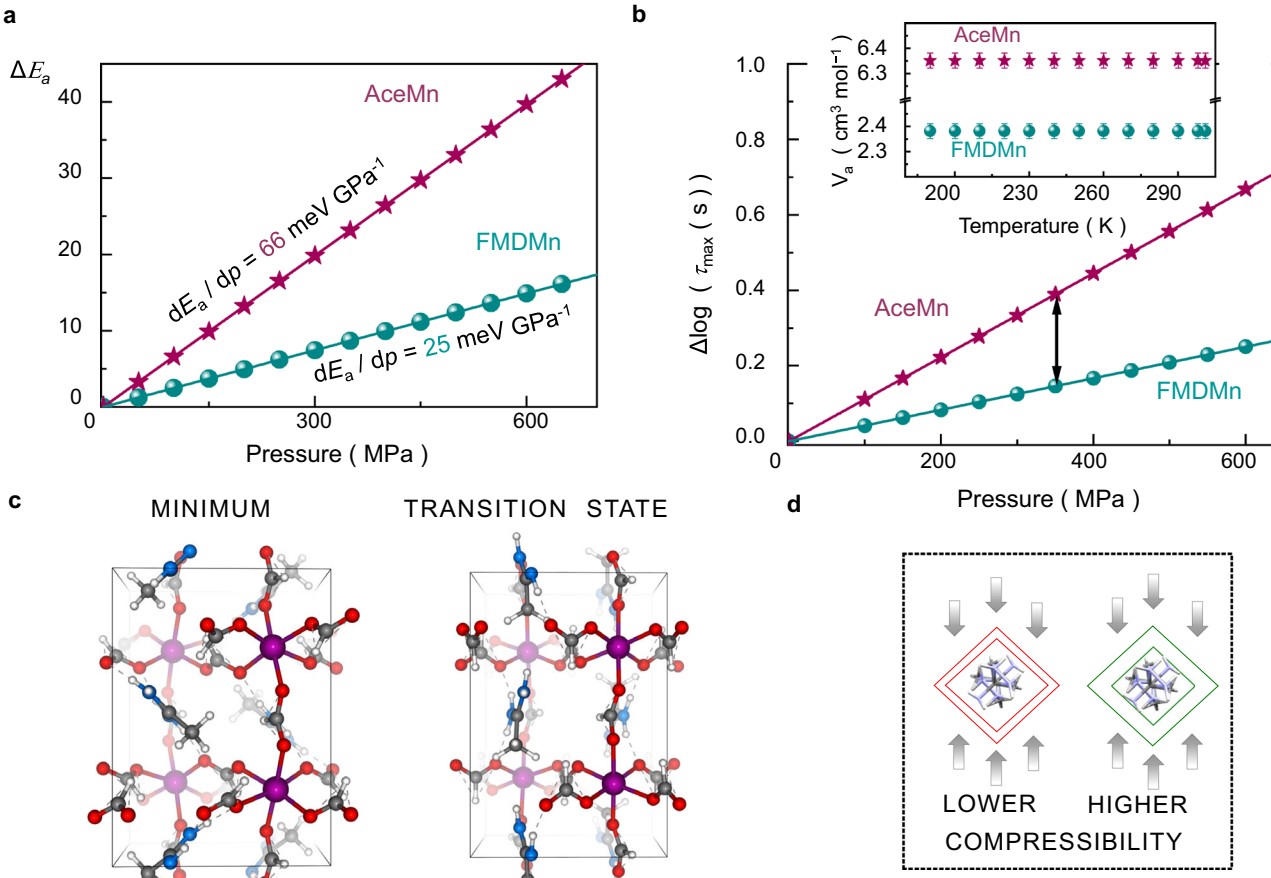

**Fig. 6 | Volume-related effects in relaxation dynamics. a** Pressure-induced increase in activation energy parameter for FMDMn (blue dots) and AceMn (red stars). The colored lines represent fit curves defined by Eq. (6). **b** Pressure dependence of $\Delta\log\tau_{max}$ for FMDMn and AceMn at 298 K. Here, $\Delta\log\tau_{max}$ is calculated as changes in relaxation times $\log\tau_{max}$ relative to the ambient-pressure value of this parameter. The inset shows activation volume $V_a$ at various temperatures for FMDMn and AceMn. **c** Possible changes in the manganese-formate skeleton induced by reorientation of Ace$^+$ cations. Mn, O, C, N, and H atoms are marked in purple, red, gray, blue, and white, respectively. **d** Schematic illustration of the diversified effect of external pressure on the cage cations in hybrid MOFs due to different compressibility of the organic-inorganic framework. The metal-organic framework is schematically shown as red or green lines. In turn, the H, C, and N atoms of the cage cation are marked in white, black, and blue, respectively. Source data are provided as a Source Data file.

(refer to red open and closed points in Fig. 5d). Consequently, we can conclude that pressure and temperature exert similar effects on the crystal structure in both FMDMn and AceMn. In contrast, the dielectric relaxation proves to be far more sensitive to temperature changes. This becomes evident when plotting $\tau_{max}$, calculated from Eq. (5) based on the known pressure–temperature, versus the experimentally determined unit-cell volume for FMDMn and AceMn (Fig. 5e, f). As presented in Fig. 5e, f, volume contraction has a significantly smaller impact on relaxation times compared to temperature effects for both FMDMn and AceMn. This observation emphasizes the negligible influence of thermal expansion on the energy barrier associated with the cage cations' movements within the cavities. Consequently, these findings provide an explanation for the activation-like Arrhenius behavior observed in the dependence of $\tau_{max}(T)$ observed in AceMn and FMDMn, with temperature-independent parameter $E_a$. They also emphasize that the volumetric changes and related shifts in the strength of hydrogen bonds or other interactions do not constitute the sole determinant governing the relaxation dynamics in HOIPs.

According to formula (5), $\tau_{max}$ scales with the activation volume parameter in AceMn, FMDMn suggesting a similar scaling also applies to other HOIPs. The activation volume is an important parameter in the equation of state as it determines the pressure-induced changes in relaxation times and the corresponding shift of the relaxation peak[51]. It also determines the magnitude of pressure-induced changes in $E_a$.

Namely, by combining Eq. (5) and a classical Arrhenius law, one can derive:

$$E_a(p) = E_{a(0MPa)} + pV_a \tag{6}$$

As presented in Fig. 6a, if $V_a$ is positive, $E_a$ increases with pressure in a linear manner and the slope coefficient of the $E_a(p)$ dependence is equal to $V_a$:

$$\frac{dE_a}{dp} = V_a \tag{7}$$

These formulas effectively explain the experimentally confirmed absence of pressure-induced increase of $E_a$ in FMDMn despite the electric-field-induced reorientation of the FMD$^+$ cations[12]. For this material, Eq. (6) predicts a small increase of $E_a$ at high pressure by 25 meV GPa$^{-1}$, resulting in an increment of about 8 meV when applying 300 MPa (or 16 meV when applying 600 MPa), which fits within the experimental uncertainty of 20 meV[12]. In turn, for AceMn, $E_a$ increases under pressure by 66 meV GPa$^{-1}$ (equivalent to 20 meV under 300 MPa pressure or 40 meV under 600 MPa), which was confirmed experimentally[13]. It is thus evident that the pressure-induced changes in $E_a$ increase with rising $V_a$ value. Similarly, at constant temperature,

$\tau_{max}$ increases under pressure with rising $V_a$:

$$\ln\tau_{max}(p) - \ln\tau_{max}(0.1 MPa) = \Delta\ln\tau_{max} = \frac{\Delta p V_a}{RT} \qquad (8)$$

This rule is well illustrated by the $\Delta\log\tau_{max}(p)$ dependencies for the studied hybrid perovskites FMDMn and AceMn (Fig. 6b). It can be shown that $V_a$ is a temperature-independent material constant if the relaxation dynamics is described by the equation of state given by formula 5 (see inset in Fig. 6b). This parameter does not depend on the size of reorienting cage cation. Our DFT investigations indicate that the ratio of the $Ace^+$ and $FMD^+$ cation sizes (defined as the ratio between their van der Waals volumes o $V_{Ace^+}^{vdW}/V_{FMD^+}^{vdW}$) equals 1.7. In contrast, experiments show that the ratio of the $V_a$ parameters for AceMn and FMDMn is equal to $V_{a,AceMn}/V_{a,FMDMn} = 2.7$. Furthermore, the relaxation of larger 1,4-butanediammonium cations (DAB$^{2+}$, [NH$_3$-CH$_2$-CH$_2$-CH$_2$-CH$_2$-NH$_3$]$^{2+}$) within the zinc-formate cages is related to much smaller $V_a$ compared to FMDMn and AceMn[14]. In order to define the activation volume, we study the exemplary reorientation path of $Ace^+$ cations within the $P2_1/n$ phase, employing DFT solid-state nudged elastic band calculations.

According to previous studies on FMDMn and AceMn, the low-temperature motion of the cage cations can mirror the mechanism typical for the HT phase[12,13]. Consequently, we focus on rotating the $Ace^+$ cation around its C−N bond in our simulations. As shown in Fig. 6c, such transformations reshape the organic-inorganic cages and induce variations in their volume. The transient state is not only a higher-energy structure but also features larger cavity sizes. The simulated volumetric barrier for the reorientation of a single $Ace^+$ cation within phase II is roughly 7.5 Å$^3$ (see Supplementary Fig. 9 for more details). This value corresponds reasonably to the experimentally determined $V_a$ parameter for AceMn (6.6 ± 0.1 cm$^3$ mol$^{-1}$), equal to 11 Å$^3$ per reorienting unit. The discrepancy between theory and experiment may originate from the chosen calculation methodology, which only considers one reorientation possibility out of many possible field-induced motion schemes. Indeed, prior studies on DABZn suggested $V_a$ to be a direction-dependent (and, consequently, mechanism-dependent) parameter[14]. Hence, based on the DFT calculations, we define activation volume as a parameter delineating the volumetric barrier for the relaxation of cage cations within the cavities. This definition mirrors that derived for other systems[53–56]. However, in HOIPs, $V_a$ depends on numerous factors, including cage cation flexibility, mechanism of its motion, or rigidity of the entire cavities. As illustrated schematically in Fig. 6d, increased rigidity of the metal-organic framework leads to smaller volume contraction of the cages during isothermal compression, and the reorienting cage cation is less affected by external forces. As a result, the volumetric contribution to the relaxation times is reduced, so $\tau_{max}$ changes less with pressure. This decrease in pressure sensitivity, according to Eq. (8), lowers the $V_a$ value. Furthermore, the increased flexibility of the cage cation helps it adapt better to pressure-induced changes in the cavities, enhancing this effect.

Finally, defining $V_a$ as a volumetric barrier for the relaxation process also explains the significantly larger influence of temperature on $\tau_{max}$ (see Supplementary Note 2 for detailed mathematical formalism). At high temperatures, the metal-organic framework possesses enough thermal energy to adjust and accommodate the reorientations of cage cations even under high pressure. Their structural flexibility decreases significantly with lowering temperature, resulting in a dramatic slowdown of the relaxation dynamics of the cage cation. The non-equivalent pressure and temperature effects, previously reported for methylammonium lead iodide (MAPbI$_3$)[57], explain well why the FMD$^+$ cations do not order under isochronous conditions.

In conclusion, ambient- and high-pressure X-ray diffraction, Raman scattering, and broadband dielectric studies were performed on model formamidinium manganese(II) formate (FMDMn) and acet-amidinium manganese(II) formate (AceMn) to expand our knowledge on high-pressure behavior of hybrid formates and derive fundamental principles governing their relaxation dynamics under various pressure regimes. Firstly, the AceMn crystals exhibit a phase transition at 1 GPa from ambient-pressure phase II (space group $P2_1/n$) to high-pressure orthorhombic phase III (space group $Pbca$). Phase III of AceMn can undergo astonishing overcompression above 5 GPa, exceeding 2.3 GPa, coexisting with an amorphous solid and contributing to a mosaic-like topology in this metastable region. Secondly, the compression of FMDMn leads to a reduction of the cage volume followed by the ordering of cage cations, without altering the symmetry nor considerably increasing the activation energy. This mechanism is associated with reorientation of the cage cations and structural adjustments of the framework. Consequently, the relaxation dynamics show two significant contributions (thermal and volumetric), with thermal energy and thermally-activated librations within the crystal mostly affecting the relaxation times. As a result, relaxation times do not scale with unit-cell volume or the strength of hydrogen bonds despite similar pressure and temperature evolution of the unit-cell parameters. This observation is reflected in a small activation volume parameter, redefined for HOIPs as a temperature-independent material constant. This parameter determines the volumetric barrier for relaxation processes and depends on cage cation flexibility, the mechanism of its motion, and the rigidity of the entire cavities. Hence, the results indicate a complex relationship between structure and dielectric response in HOIPs under pressure.

## Methods
### Materials
The subject of this research is crystalline compounds FMDMn and AceMn. Both compounds were synthesized as small crystals according to the procedures utilized by us before[21,22]. In order to obtain FMDMn, a mixture of cyclobutane-1,1'-dicarboxylic acid (4.2 mmol), MnCl$_2$ (8.4 mmol), and formamide (50 mL) was heated at 130 °C for 24 h in a Teflon-lined microwave autoclave. Light pink crystals were collected after slow overnight cooling. They were subsequently washed with ethanol (3 × 5 mL) and finally dried at room temperature. In order to synthesize AceMn, acetamidine hydrochloride (40 mmol) was dissolved in methanol (15 mL) and the resulting solution was mixed with formic acid (80 mmol) and trimethylamine (10 mmol). The mixture was placed at the bottom of a plastic vial, which was followed by the careful addition of 10 mL of a methanol solution of manganese(II) perchlorate hydrate (2 mmol). The vial was sealed and left undisturbed for 48 h. The formed crystals of AceMn were subsequently harvested, washed with methanol (3 × 5 mL), and dried at room temperature.

### Ambient- and high-pressure broadband dielectric spectroscopy
Ambient-pressure dielectric measurements of FMDMn and AceMn were conducted on polycrystalline and pelletized samples with a diameter of 5 mm. Prior to measurements, parallel planes of the samples were painted with silver paste to ensure optimal electric contact and thoroughly dried in a vacuum oven at 330 K. A Novocontrol Alpha impedance analyzer was utilized for the broadband dielectric investigations. During measurements, an ac current in the frequency range of $10^{-1}$–$10^6$ Hz was applied across the samples. For both compounds, dielectric spectra were collected every 2 K between 170 and 320 K under quasi-static conditions. For this purpose, the temperature was stabilized prior to each spectrum collection with nitrogen gas. The temperature was controlled by a Novocontrol Quattro system with a precision exceeding 0.1 K.

The same samples of FMDMn and AceMn were then utilized for high-pressure dielectric measurements. However, in this case, they were glued with a silver paste to the high-pressure capacitor, after which they were subjected to drying in a vacuum oven at 330 K for

24 h. Each so-prepared high-pressure capacitor was subsequently enclosed in a Teflon capsule filled with Julabo Thermal HL90 silicon oil and placed inside an LC20 T-type high-pressure chamber (designed by UNIPRESS for hydrostatic pressure up to 1.8 GPa). The whole high-pressure system was put into a thermostatic mantle and placed on a hydraulic press. High-pressure dielectric measurements were conducted during compression under isothermal conditions of 298 K. During measurements, the temperature was stabilized and controlled by Huber Tango Unistat thermostatic bath with a precision exceeding 0.1 K. Ambient-pressure alike, and dielectric spectra were recorded by a Novocontrol Alpha impedance analyzer. During measurements, an ac current in the frequency range of $10^{-1}$–$10^6$ Hz was applied across the samples. Prior to each spectrum collection, pressure was stabilized by at least 15 min to ensure quasi-static conditions and the measurements were performed every 50 MPa up to 1500 MPa for FMDMn and (due to technical reasons) up to 1000 MPa for AceMn.

### High-pressure single-crystal X-ray diffraction
The high-pressure experiments were performed with a modified Merrill-Bassett DAC cell[58] with diamond anvils supported on steel disks. The diamond culets were 0.8 mm in diameter. The gasket was made of 0.1 mm thick tungsten foil with sparked-eroded holes of 0.4 to 0.5 mm in diameter. The pressure was calibrated using the ruby fluorescence method[59,60], and Daphne oil 7575 was used as the pressure-transmitting medium. Single-crystal X-ray diffraction measurements were performed using a 4-circle diffractometer equipped with a CCD detector and MoKα X-ray source (λ = 0.71073 Å). The DAC chamber was centered using the gasket-shadowing method[61]. The CrysAlisPro software was used for the data collection and processing. All the structures were solved with direct methods using Shelxs and refined with Shelxl using the Olex2 suite[62,63]. Structural drawings were prepared with program Mercury CSD 3.8[64].

### High-pressure Raman scattering
High-pressure Raman spectroscopy was performed using a LabRAM HR Evolution spectrometer (from Horiba), set to a resolution of 2 cm$^{-1}$, and equipped with a thermoelectrically cooled CCD detection system. The spectra were collected with a 20× long working distance Olympus objective lens and the excitation was provided by a 514.4 nm argon laser line in backscattering geometry, with a power of 2.0 mW focused onto the sample. A diamond anvil cell (DAC) SS Syntek design (manufactured by NOVARETI LTDA) was employed to achieve high pressures. Mineral oil was used as the pressure-transmitting medium. The sample was loaded into a 100 μm diameter hole drilled in a 200 μm thick stainless-steel gasket using an electric discharge machine from Almax easyLab. The pressure was determined by monitoring the shifts of the ruby R1 and R2 fluorescence lines.

### Density functional theory
Density functional theory[65] was used to model and analyze the structure of FMD$^+$ and Ace$^+$ cations. Their van der Waals volumes in a single-molecule approach by means of B3LYP hybrid density functional[66–68] combined with Pople's 6-311++G(d,p) basis set[69]. These calculations were carried out using the Gaussian 16C.01 program package[70].

To study relaxation dynamics and pressure-induced phase transition in AceMn, periodic DFT simulations were conducted using projector-augmented wave (PAW) pseudopotentials within the Vienna Ab initio Simulation Package (VASP, version 5.4.4)[71–73]. The simulations employed the Perdew-Burke-Ernzerhof (PBE) exchange-correlation functional[74], enhanced with D3(BJ) dispersion correction[75,76] for a more accurate treatment of van der Waals interactions, and a plane-wave cutoff energy of 500 eV. Our computational models for both phases consisted of a unit cell containing 8 Mn atoms. We utilized a gamma-centered 2 × 2 × 2 k-point grid to ensure accurate Brillouin zone sampling. Considering the presence of unpaired electrons on d-orbitals of

Mn atoms, spin polarization was incorporated, adopting a high-spin ferromagnetic configuration with a magnetization value of 5 μ$_B$ per Mn atom. To mitigate self-interaction errors, a Hubbard U correction (U = 3.8 eV[77]) was applied to the d-orbitals of Mn. The convergence criteria for electronic steps and geometry optimization were set to $10^{-6}$ eV and 0.01 eV/Å, respectively. Volume-optimized energy calculations were performed under constant-volume conditions, allowing for variations in lattice parameters (ISIF = 4). Phonon calculations were carried out using Phonopy (version 2.19.1) and the Parlinski–Li–Kawazoe method[78,79], with finite difference single point calculations in a 1 × 1 × 1 supercell. Phonon frequencies were determined using a 14 × 14 × 14 sampling mesh for thermodynamic functions and a 4 × 4 × 4 mesh for the phonon density of states (detailed convergence tests are provided in Supplementary Fig. 10). Pore size distribution was analyzed with PoreBlazer software[80]. The minimum energy path for acetamidine rotation was identified using the solid-state nudge elastic band (SS-NEB) method[81], facilitated by the Transition State Atomistic Simulation Environment (TSASE), an extension of the Atomic Simulation Environment (ASE)[82].

## Data availability
Data that support this research are available in its Supplementary Information files. The source dielectric, Raman, and structural data generated in this study are provided in the Source Data file. All other data are available from the corresponding author upon request. The X-ray crystallographic coordinates for the structures of AceMn and FMDMn reported in this study have been deposited at the Cambridge Crystallographic Data Centre (CCDC), under deposition numbers 2287942-2287953 for FMDMn and 2287917-2287940 for AceMn. These data can be obtained free of charge from The Cambridge Crystallographic Data Centre via www.ccdc.cam.ac.uk/data_request/cif. Source data are provided with this paper.

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

## Acknowledgements

F.F. gratefully acknowledges Polish high-performance computing infrastructure PLGrid (HPC Centers: ACK Cyfronet AGH) for providing computer facilities and support within computational grant no. PLG/2023/016135.

## Author contributions

Conceptualization: A.N., A.K. and A.S.; Sample preparation: M.M.; Experimental investigation: S.S., K.R., A.K., A.N., A.Z.S., A.J.B.S. and W.P.; Theoretical investigation: A.N. and F.F.; Project administration: A.S.; Supervision: A.S., A.K., S.P., and M.M.; Visualization: A.N., S.S. and A.Z.S.; Writing: A.N. and S.S.; Review & editing: all authors.

## Competing interests
The authors declare no competing interests.

## Additional information

[1]Laboratoire National des Champs Magnétiques Intenses, EMFL, CNRS UPR 3228, Université Toulouse, INSA-T, Toulouse, France. [2]Department of Experimental Physics, Wrocław University of Science and Technology, Wrocław, Poland. [3]Faculty of Chemistry, Adam Mickiewicz University, Poznań, Poznań, Poland. [4]August Chełkowski Institute of Physics, University of Silesia in Katowice, Chorzów, Poland. [5]Institute of Low Temperature and Structure Research, Polish Academy of Sciences, Wrocław, Poland. [6]Department of Chemical and Biological Engineering, Northwestern University, Evanston, IL, USA. [7]Department of Micro, Nano and Bioprocess Engineering, Wrocław University of Science and Technology, Wrocław, Poland. [8]Faculdade de Fisica, Universidade Federal do Para, Belem, Brazil. ✉e-mail: andrzej.nowok@pwr.edu.pl; katran@amu.edu.pl; adam.sieradzki@pwr.edu.pl

