## [Peer Review File · Nature Communications]

Temperature and volumetric effects on structural and dielectric properties of hybrid perovskitesREVIEWER COMMENTS

Reviewer #1 (Remarks to the Author):

In the manuscript titled "Pressure-Driven Paradigm Shift: Unveiling Activation Volume Effects in Hybrid Organic-Inorganic Compounds", the focus is put on the high-pressure behavior of the perovskite materials AceMn and FMDFM. High-pressure single crystal X-ray diffraction studies and high-pressure broadband dielectric spectroscopy were performed. It is found that, counterintuitively, AceMn exhibits a high-pressure induced phase transition that comes with an increase of symmetry. Additionally, analysis of the dielectric spectroscopy data shows that activation volumes can be seen as material constants, which includes all parameter that contribute to pressure induced relaxation dynamics of the A-cation in one constant.

The most interesting finding is by far the symmetry-increase upon pressure increase. Whilst I cannot judge on the quality of the crystal structure solution, I simply didn't find any crystallographic data, I trust the authors in their analysis; however, no explanation nor theory is given why this very unusual behavior occurs. The second point that is covered in the manuscript is an interesting, very detailed analysis on the relaxation dynamics of the A-site cation, to my knowledge one of rare cases of such an in-depth analysis. The broader impact of having a defined a parameter such as an activation volume is broadly unclear though, especially given that chemical intuition tells you exactly this. The authors themselves mention that V_a is a complex material constant, and the manuscript falls short in highlighting its broader significance. The latter, however, might also be related to a very convoluted way of how data and discussion are presented. Additionally, there are *many* smaller points, such as imprecise or overenthusiastic phrasing, which makes everything harder to digest, starting, for instance, with the title. Overall, I think these are nice results, with very detailed analysis of dielectric data. Without a strong theory, however, why the unusual high-pressure structural behavior is observed I cannot recommend publication (see point A). I would be happy to re-review a heavily revised version of the manuscript.

A)The symmetry-increase upon increasing pressure for AceMn is intriguing and I agree with the authors that normally a symmetry decrease is expected which enables efficient strain reduction. Based on this interpretation, it seems another parameter plays a role that outweighs the energy penalty due to strain contributions. Previously it was shown that coupling of multipoles might be such a parameter, but I believe won't play a role here. Do the authors have a theory for the observed high-pressure phase behavior?

B)The merit of the whole paragraph discussing the isochronous line of relaxation (p.12) up to p.15 (top) seems overly complicated and the significance is broadly unclear. On p.15 it is even mentioned by the authors that this behavior is expected upon qualitatively arguments based on Figure 3c (see p. 15 bottom). This whole part should be simplified, and main conclusions clearly highlighted. For instance, it would be a good start to put the motivation for the whole discussion & analysis at the very beginning.

a)I cannot see any paradigm-shift, please rephrase the title.

b)The intro mentions many pressure-related effects for semiconducting HOIPs, while the presented materials are isolators. Due to the difference in chemistry, the ramifications of these on the HOIPs is unclear at minimum, and it is not discussed if conclusion drawn from analytic data on V_a can be mapped onto semiconducting HOIPs. Therefore, placing them dominantly in the introduction is heavily misleading. A concise introduction into formate-based perovskites or similar materials should be provided instead. Alternatively, I could see how showing the significance of V_a for semiconducting HOIPs would result in only a minor change of the introduction.

c)P. 5, top. Stating that "Resulting in almost no compressibility" for FMDMn with a value of β_{ax} of 3.38 GPa⁻¹ is confusing. Even the FMDMn compressibility is a magnitude smaller (-4.3 TPa⁻¹). This makes the hydrogen-bonded interpretation obsolete too. Please correct.

d)Looking through the principal axis analysis of compressibilities that is given in the supporting information, AceMn shows no NLC behaviour while FMDMn exhibits NLC along [10-2], i.e. pressure induced changes of the unit cell can be related to a large change of β . This should be properly included in the discussion, see 10.1107/S0021889812043026 for an example where a wine-rack type movement is discussed. Currently, the discussion on compressibilities that is given in the paper is on the borderline of wrong.

- e) Following point b) and c) I suggest putting a Table into the manuscript that provides an overview of lattice parameters changes / strains / compressibilities for both compounds.
- f) NLC is, very much like NTE, not a rare phenomenon anymore (10.1039/C5CP00442J), counterintuitive maybe (p. 7)
- g) Related to the high-pressure phase transition of AcMn , "a collapse of all three voids" is mentioned. Which voids are we talking about? This is unclear.
- h) The sentences "Interestingly, the electric-field induced reorientations of FMD+ cations in low-temperature phase remains unchanged under isochoric conditions. It occurs at 110 K at 0.1 MPa and at approx. 1 GPa at 298 K, corresponding to unit-cell volume of 865.0(7) and 824.6(18) Å³, respectively." does not make any sense, which is the underlying isochoric process? Also, it is unclear where the data is coming from.
- i) The y-axis labelling of the insets in Figure 3a and b are wrong.
- j) I believe in deriving eq. 5 isothermal conditions are assumed, this should be added since eq. 4 is a function of p and T.
- k) P. 16 "Va is complex temperature-independent material constant" – this is a generic sentence that arguably will be true for all HOIPs. At the same time, comparing the van-der-waal volume ratio with the ratio of Va for both compounds underestimates the whole underlying complexity. The only scenario I see where both values would agree is if both A-cations would be spheres. A more thorough discussion on the interpretation of Va based on chemical arguments should be given.

Reviewer #2 (Remarks to the Author):

The authors investigate the phase transition under different pressures. And they discussed the coupling motion in phase transition and revealed its effect in dielectric relaxation phenomenon. I must say sorry that these discussions are too trivial and some views are unclear. Therefore, I can not recommend this paper for publication, in particular to Nature Commu.

1. The studies of coupling motion between A-site cations and inorganic framework have been investigated under low pressure (<1Gpa). The majority of the conclusions discussed by the author at high pressure are consistent with the phenomena observed in low-pressure phase transition (< 1GPa).

2. "This effect, of negative linear compressibility (NLC), is a rare phenomenon among three-dimensional HOIPs". In this paper, the author mentions the presence of the NLC phenomenon in the model systems, but there is a lack of thorough discussion regarding the relationship between this rare phenomenon at high pressure and the properties of material.

3. "Finally, our results provide compelling mathematical evidence that pressure-induced phase transitions in HOIPs are not only governed by the dynamics of movement of A-site cage cations, but it is a combined effect of ..." We think the control of pressure is an effective way to tune the volume of HOIPs. Further, the cage size and the cation reorientation have a strong coupling by H-bond. So that the volume can significantly control the orientation of A-site cations. We think this conclusion is trivial and have been discussed in previous work. Maybe, in-depth theory are needed to revealed the coupling relationship.

4. The pressure will most probably enhance the coupling of cation and framework. Some theoretical papers for coupling motion in HOIPs here: [Phys. Rev. B 94, 045201], [Phys. Rev. Lett. 122, 225701], [Phys. Rev. B 100, 09410].

Reviewer #3 (Remarks to the Author):

Key results

The paper by Nowok et al. describes the dependence of dielectric relaxation of two organic-inorganic hybrid materials, namely acetamidinium manganese(II) formate (AceMn) and formamidinium manganese(II) formate (FMDMn). The authors previously published a similar study on acetamidinium manganese(II) formate (AceMn) alone. None of the actual represented data are the same, but the spirit is analogous.

High hydrostatic compressive pressure changes the unit cell (volume). This also changes the dynamics of the contained organic molecules. This is measured as macroscopic dielectric data. In particular, the dielectric losses represented as ϵ'' are used to describe the behavior.

The data are given up to 700 MPa applied pressure. In one inset the unit cell volume is monitored up to 2.5 GPa.

At a certain pressure value, a phase transition is induced.

The main result is that the dielectric response is more susceptible to temperature than to pressure indicating that the molecular dynamics yield a high contribution to the dielectric response. The working out of this major finding within the manuscript is not well represented in the text.

Validity

All data appear fully valid.

Significance

It is very interesting and valuable to understand the dynamics of (mostly small) organic molecules in hybrid organic inorganic crystals. This dynamic response can explain optical and electrical effects. The high (hydrostatic) pressure data are thus valuable and significant.

In a prior paper a similar approach was taken by the same authors, but only on one of the two compositions was studied. In the earlier paper, temperature was varied for a few given pressures, here pressure is varied for room temperature.

In the present paper both compositions are compared and the pressure dependent dataset is completed.

Data and methodology

All data appear sound and reproducible. There is no doubt here.

Analytical approach

The data analysis is clear and sound. Only the interpretation is written in a confusing manner. It is overall still too technical and not aimed at working out the main conclusions of the paper.

Suggested improvements

Your suggestions for additional experiments or data that could help strengthen the work and make it suitable for publication in the journal. Suggestions should be limited to the present scope of the manuscript; that is, they should only include what can be reasonably addressed in a revision and exclude what would significantly change the scope of the work. The editor will assess all the suggestions received and provide additional guidance to the authors.

Clarity and context

The text needs certain improvement in the English. It is rather the willingness to produce beautiful English which then generates wordings that are not appropriate or vocabulary that is not correctly used. In this context there is much improvement potential.

The introduction should also tell why exactly these two compounds are compared, what are the expectations for comparing these two and how the experiment confirms these expectations or necessitates a different interpretation.

References

Are correct.

Your expertise

Matches the content of the paper.

Providing constructive feedback

the best and most constructive reports suggest specific improvements; such feedback can be used by authors to improve their manuscript to the point where it might be suitable for acceptance. Even in instances where manuscripts are rejected, your report will help authors interpret the editor's decision and improve their work prior to submission elsewhere.

You should be direct in your report, but you should also maintain a respectful tone. As a matter of policy, we do not censor the content of reviewer reports; any comments that were intended for the authors are transmitted, regardless of what we may think of the content. On rare occasions, we may edit a report to remove offensive language or comments that reveal confidential information about other matters.

Confidential comments to editors

Your comments to the authors should contain all feedback pertaining to the scientific evaluation of the manuscript, as detailed above. Confidential comments to the editor may be the appropriate

place to discuss sensitive information or opinions, including any potential ethical issues with the work, or information that cannot be shared with other reviewers, such as any previous interaction with the manuscript at another journal, but should in no way contradict the comments to the authors.

Technical comments:

Overall, the manuscript needs an English fine polishing. This was actually hindering the view for the content and the review of the document.

Overall, I had the impression that the intention was to produce well worded English, but the words were sought from the wrong sources and are actually not suitable for the intended meaning. I think many of the detailed comments later are simply due to this effect.

Abstract:

The abstract is written in report form, “we have done ...”. It should actually only contain the essentials of the major content of the paper and a scientific result summary.

Title:

I would omit the word “unveiling” in the title, it is meaningless towards the paper content. So I suggest: “Pressure-Driven Paradigm Shift: Activation Volume Effects in Hybrid Organic-Inorganic Compounds”

Introduction:

The authors are very familiar with the chemical formulae of the organic molecules, for the new reader at least the sum formula for both compounds must be given early on in the introduction and an image would always be helpful.

What do the authors mean by “phase over-compression”. This seems to make no sense here. I think it is an issue of English language.

There is discussion on “A-site cage ions or their components“ straight in the introduction without introducing the material that is actually concerned. This is confusing. We have 3 ions, so what is the A-site. Also these structures are more complex than simple perovskites, so site assignment is not trivial.

“phenomenon” is the singular form, “phenomena” the plural, so write

I personally do not like this sentence and similar statements when they do not apply:

“Our results offer valuable insights for future HOIP research and the design of electronic and optoelectronic devices.”

3 GPa is a very high pressure and not available in any realistic device structure or industrial process. Thus, I would keep the argument along the gained scientific knowledge rather than technological impact of the findings. I do not deny that this material can be employed in technology, but not under high pressure.

“plethora” is a very strong word and exaggerates the content of the paper.

Results:

The temperature/pressure line in Table I is confusing.

This sentence is strange: “righting the crystal direction [010] along which the H-bonds are aligned“

The authors state that there are MnO₆-octahedra which rotate, where are these in this crystal structure or what do the authors mean?

Page 6: “This phase experiences a monotonic compression of the unit cell with increasing external pressure“. Do the authors consistently mean hydrostatic pressure? This is nowhere stated in the manuscript.

What is “its ‘wine-rack’-type strain“, this is not established terminology and also not self-explaining.

The term “negative linear compressibility” is rather misleading. “Negative compressibility” under hydrostatic compression can only occur, if the crystal tilts. Thus, the seen effect is rather a shear effect and not negative compressibility which itself can only occur at a phase transition.

The authors use the term “void”. In crystallography this is defined as “Voids — small regions where there are no atoms, and which can be thought of as clusters of vacancies”. As far as I understand the sentence, this is not what the authors mean.

I think this statement is wrong: “In most cases, symmetry of molecular crystals is reduced in order to increase the number of degrees of freedom in structure, so it efficiently reduces the strain associated with the compression.” Which degrees of freedom are there supposed to be more? I think the term “degree of freedom” is not what the authors mean.

Figure 2: the labels (d) and (e) are not mentioned in the figure caption.

What do the authors mean by “ambient-pressure H-bonding pattern“ ? Do you mean the special arrangement of the H-bonds? If so, please write so.

What do the authors mean by “collapses about the cations”.

Again MnO₆-octahedra are mentioned. This is somehow very unclear. If Mn borrows two oxygen from each of the formate ions for the crystal structure, then this must be shown in a detailed image of the 3D structure. The crystal structure as shown in Figure 2 does not display how the formate ions order in the 3D structure. This was much better solved in the Physical Chemistry C-paper by the same authors.

English: “progressing over a period of 2 weeks” should be after “reflection” in the same sentence.

This sentence is in “Results” but to me it means a lot of interpretation: “The increased unit-cell dimension of residual phase III by ca. 1.3 % suggests a mosaic topology of amorphous higher-density regions.” It should be discussed in “Discussion”.

English: write: “This relaxation manifests itself as a bell-shaped ...“

How can you assume “under isochoric conditions” ? These materials undergo very high pressure loads, why should the volume not change? I do not understand this. You even gave compressibility coefficients.

What do the authors mean by “due to contraction of the organic inorganic caverns”? Do they mean unoccupied space in the unit cell? This is unusual wording. A cavern is essentially a grotto. So I think this word is simply wrong.

Replace “elongate” by “extend”. Elongate is a word that applies to space, not to time.

I do not understand the meaning of this sentence: “determines the p - T isochronous line of relaxation times”. What happens here at the same time?

This sentence: “while $\tau_{Max}(p)$ remains constant under isochronous conditions“. That something stays constant in a single moment is trivially true.

The following sentence is also not understandable: “that undergo isochronously, i.e., which are controlled by the dynamics of reorienting moieties “. As the word isochrones is used much in the manuscript and it is a rare word and very specific, it must be clearly defined how it is used in the manuscript. One could e.g. use equation 6 to define such lines / times, but this must be done very clearly. This could be provided in the supplementary material, but it must then also be referenced in the main manuscript.

I do not understand this conclusion: “However, provided that the dielectric relaxation phenomena in such HOIPs are connected to the electric-field induced reorientation of the organic A-site cage cations, it can be concluded that the observed pressure-induced phase transitions in these materials (e.g., AceMn) cannot be governed solely by the dynamics of cage cations.” This is now a scientific matter, not English. The dielectric response contains all contributions, namely cage motion, molecular rotation and relative shift of the positively and negatively charged sub-lattices. It is only a question in which frequency ranges these respective mechanisms become dominant in the dielectric response.

There are no red points in Fig. 4 (c). There is black and orange.

“According to the calculations made for the reference τ_{Max} equal to 1 μ s based on the equations of state, the elevation of pressure up to 300 MPa shifts the relaxation peak as much as a decrease in temperature by about 4.5 K for FMDMn and 11 K for AceMn (see Fig. S2).” This statement is rather surprising. I urge the authors to reproduce the data / check the calculations to make sure that this statement is correct.

“ V_a is a complex temperature-independent material constant.“ What is so complex about a constant? If something is constant to me it appears more simple than something that changes.

The summary is too long (two pages). It should be reduced to the essentials that are found in the paper and not summarize everything. A conclusion is more appropriate here than a summary.

“In summary, ambient- and high-pressure X-ray diffraction and broadband dielectric studies were performed on model formamidinium manganese(II) formate (FMDMn) and acetamidinium manganese(II) formate (AceMn) to unveil their exceptional features and derive fundamental principles governing the behavior of these prototypical HOIPs under various pressure regimes.” This sentence is completely meaningless in a summary. The reader does not want to read how important the work is, but wants to understand from the data what is significant. The summary should be fully content based and not blabla.

“overcompression” is a word that must be clearly defined or eliminated from the text. It is not understandable.

“Moreover, V_a is a complex temperature-independent material constant for AceMn and FMDMn. It not only governs the volume requirements for the movement of the cage A-site cations in response to external stimuli but also incorporates information about the stiffness of the cavities.” How does it “incorporate” information about stiffness? This is not explained in the manuscript.

“Finally, our results provide compelling mathematical evidence that pressure-induced phase transitions in HOIPs are not only governed by the dynamics of movement of A-site cage cations, but it is a combined effect of appearing...”. I do not see any mathematical evidence in this manuscript. It is all experimental with a few fit functions. There are no mathematical derivations, it is all thermodynamics.

“This finding opens up new avenues for research into the dynamic behaviour of A-site cage cations under pressure, urging further exploration of the underlying physics in these materials. Consequently, our study unravels features of AceMn and FMDMn and establishes fundamental principles governing the behaviour of formate-based HOIPs under varying pressure regimes.” Two completely meaningless sentences. Sentences like this one must be removed they only unnecessarily extend the paper and thus make it unreadable. Only concise clear texts are pleasant to read.

Reviewer #4 (Remarks to the Author):

The manuscript describes a thorough reexamination on the effect of pressure on the structural behavior of two HOIPS – formadinium manganese formate, and acetamidinium manganese formate.

Fundamentally, I believe this work presents careful data analysis that is of interest to the broad readership of Nature Communications. I believe their interpretation of the single crystal/dielectric data is good and provides a useful metric for examining these intriguing compounds. However, there are areas of the manuscript that need addressing. Issues are predominantly the presentation of data, but there is also a potentially glaring gloss over of Raman data mentioned. Without that data included, though the SC/dielectric data presented is the main focus of the work, the manuscript likely needs another round of examination.

Most important:

Figure S4 of the Raman data is mentioned, but not included in the manuscript. I am uncertain if the authors are referencing Raman experiments formerly published that they are including in this manuscript, or if there are new Raman experiments that confirm or align with that previously published data / the authors own crystallographic/dielectric observations. I believe if the authors are referencing and analytically leaning on the Raman data, that this data needs to not only be included in the SI, but be more thoroughly presented/analyzed and referenced. I believe that the authors own concluding paragraph, which suggests the impacts of "vibrations" may be rooted in this spectroscopy, which is currently only nodded to.

More minor thoughts:

Figure 3 needs to be reworked as currently the insets are quite difficult to read. For example, in Figure 3a, the caption refers to the open stars for calculated pressure dependence, but without excessive zooming in, the symbols are very hard to see relative to the presented data. This could be a combination of the quality of PDF provided to the reviewers, but I believe may also speak to potential what the final reader may encounter. Along those lines, Figure 4 also needs addressing, as it is unclear which way data was taken in panels a and b. I believe from the text that we're moving right to left, but this should be inherent in the figure. In general, I would suggest that the reviewers go back and look at all figures and evaluate if data is presented in a way that reflects what it should to the reader without digging.

I thank the reviewers for their incredible transparency in the amount of .cifs deposited in the CCDC. However, given that the authors are depositing 10's of single crystal data sets, there should be an accompanying table in the SI relating what specific ccdc number is what specific .cif (experimental condition). This is only an issue given the sheer number of .cifs (because if there were say just 3, they would be trivial to sort through). Doing so will give the eventual readers of this manuscript the capacity to examine this data without undue burden.

There are minor typos throughout – for example, I believe the data presented in Table 1 (ratios of the Z phases), as well as minor grammatical issues scattered through the manuscript. This stems back to my comments on the figures, and the authors should take the time to polish their work and present it in a way that is commensurate with their data analysis.

RESPONSE TO REVIEWERS' COMMENTS

Dear Reviewers, we are very thankful for the reviews and valuable comments which helped us improve the manuscript. The corrections were made according to all the comments. Please note that all changes introduced to our manuscript are marked **in red**.

ANSWERS TO REVIEWER 1

General comment

Reviewer: *“The broader impact of having a defined a parameter such as an activation volume is broadly unclear though, especially given that chemical intuition tells you exactly this. The authors themselves mention that V_a is a complex material constant, and the manuscript falls short in highlighting its broader significance. The latter, however, might also be related to a very convoluted way of how data and discussion are presented. Additionally, there are *many* smaller points, such as imprecise or overenthusiastic phrasing, which makes everything harder to digest, starting, for instance, with the title. Overall, I think these are nice results, with very detailed analysis of dielectric data. Without a strong theory, however, why the unusual high-pressure structural behavior is observed I cannot recommend publication (see point A). I would be happy to re-review a heavily revised version of the manuscript.”*

Author reply: It is unexpected to observe that the relaxation peak shifts only slightly upon isothermal compression, which is opposite to the pressure dependence of the unit cell parameters and volume. This means that the strength of hydrogen bonds and other interactions within the crystal structure does not solely govern the relaxation dynamics of cage cations. Consequently, one can tune the relaxation times (dielectric properties of the hybrid perovskites) without changing the unit cell volume and shape, just by playing with the pressure-temperature conditions. This finding changes the way of thinking about dielectric relaxations as they are commonly perceived to be strongly correlated with crystal structure. We added additional DFT calculations to explain this phenomenon in more detail. Furthermore, we added theoretical explanations for the unprecedented phase transition to the higher-symmetry structure under pressure. We also simplified the whole discussion section and conclusions.

Comment A

Reviewer: *“The symmetry-increase upon increasing pressure for $AceMn$ is intriguing and I agree with the authors that normally a symmetry decrease is expected which enables efficient strain reduction. Based on this interpretation, it seems another parameter plays a role that outweighs the energy penalty due to strain contributions. Previously it was shown that coupling*

of multipoles might be such a parameter, but I believe won't play a role here. Do the authors have a theory for the observed high-pressure phase behavior?"

Author reply: We performed additional DFT calculations to explain this phenomenon in detail. Our results are presented in the "results" section.

Comment B

Reviewer: *"The merit of the whole paragraph discussing the isochronous line of relaxation (p.12) up to p.15 (top) seems overly complicated and the significance is broadly unclear. On p.15 it is even mentioned by the authors that this behavior is expected upon qualitative arguments based on Figure 3c (see p. 15 bottom). This whole part should be simplified, and main conclusions clearly highlighted. For instance, it would be a good start to put the motivation for the whole discussion & analysis at the very beginning."*

Author reply: We agree with the Reviewer that this discussion was overcomplicated. Therefore, we removed the discussion on isochronous line from the manuscript completely. We put more attention to other surprising aspects, mentioned previously in the reply to the general comment.

Comment a

Reviewer: *"I cannot see any paradigm-shift, please rephrase the title."*

Author reply: We changed the title of the manuscript.

Comment b

Reviewer: *"The intro mentions many pressure-related effects for semiconducting HOIPs, while the presented materials are isolators. Due to the difference in chemistry, the ramifications of these on the HOIPs is unclear at minimum, and it is not discussed if conclusion drawn from analytic data on Va can be mapped onto semiconducting HOIPs. Therefore, placing them dominantly in the introduction is heavily misleading. A concise introduction into formate-based perovskites or similar materials should be provided instead. Alternatively, I could see how showing the significance of Va for semiconducting HOIPs would result in only a minor change of the introduction."*

Author reply: The introduction was rewritten. We put more focus on the formate systems.

Comment c

Reviewer: *"P. 5, top. Stating that "Resulting in almost no compressibility" for FMDMn with a value of β_x of 3.38 GPa⁻¹ is confusing. Even the FMDMn compressibility is a magnitude smaller (-4.3 TPa⁻¹). This makes the hydrogen-bonded interpretation obsolete too. Please correct."*

Author response: Thank you for your insightful comments and for pointing out the confusion regarding the compressibility of FMDMn in our manuscript. Upon reviewing our data, we have identified a typographic error in our manuscript where we reported the β_x value as 3.38 GPa⁻¹ instead of the correct value, 3.06×10^{-3} GPa⁻¹. This mistake significantly impacted the interpretation of FMDMn's compressibility. The correct value substantiates our original statement about its minimal, approaching "zero" compressibility along [010] crystal direction.

Comment d

Reviewer: *“Looking through the principal axis analysis of compressibilities that is given in the supporting information, AceMn shows no NLC behaviour while FMDMn exhibits NLC along [10-2], i.e. pressure induced changes of the unit cell can be related to a large change of beta. This should be properly included in the discussion, see 10.1107/S0021889812043026 for an example where a wine-rack type movement is discussed. Currently, the discussion on compressibilities that is given in the paper is on the borderline of wrong.”*

Author response: We fully agree with Referee 1 that the compression analysis should be properly performed by calculating the compressibility tensor, which can be then related to the crystal lattice directions and to the structure. In the revised version the compressibility tensors have been calculated by program PASCAL and applied for discussing the strain. This proper discussion based on the compressibility tensor has been added in the revised manuscript (and in the SI).

Comment e

Reviewer: *“Following point b) and c) I suggest putting a Table into the manuscript that provides an overview of lattice parameters changes / strains / compressibilities for both compounds.”*

Author response: Thank you for your valuable suggestion, we agree and believe that incorporating such a table will indeed enhance the clarity and accessibility of the data, allowing readers to readily compare and contrast the key parameters of interest. This addition aligns with our commitment to improving the quality and presentation of our research.

Comment f

Reviewer: *“NLC is, very much like NTE, not a rare phenomenon anymore (10.1039/C5CP00442J), counterintuitive maybe (p. 7)”*

Author response: Thank you for pointing out the current understanding and prevalence of Negative Linear Compressibility (NLC) in the context of our manuscript. We appreciate the reference provided (10.1039/C5CP00442J), which underscores the growing recognition of NLC in various materials. In light of this, we have revised the relevant sections of our text to reflect a more current perspective on NLC. While initially considered counterintuitive, we acknowledge that NLC has indeed become a more commonly observed phenomenon in material science. Our manuscript now includes a more detailed discussion of NLC, recognizing its increased identification and understanding in recent years. We believe this adjustment not only aligns our work with the latest scientific developments, but also provides a more comprehensive understanding of NLC in the context of our study.

Comment g

Reviewer: *“Related to the high-pressure phase transition of AceMn, “a collapse of all three voids” is mentioned. Which voids are we talking about? This is unclear.”*

Author response: Thank you for bringing to our attention the query regarding our statement: "a collapse of all three voids" in relation to the high-pressure phase transition of AceMn. We have searched our manuscript but we cannot locate the exact usage of this cited sentence. However, we presume that your comment concerns our sentence “The pressure-induced stress

and connected contraction of all free volume within the unit-cell can modify the potential well increasing in the energy barrier required for the reorientation of the Ace^+ cations.” at the end of page 15. We agree that this sentence was our unfortunate attempt to convey the interplay between the compression and dynamics of anions. We would like to point that the whole discussion part was rewritten to make it more simple to understand and that sentence was removed from the manuscript.

Comment h

Reviewer: *“The sentences “Interestingly, the electric-field induced reorientations of FMD+ cations in low-temperature phase remains unchanged under isochoric conditions. It occurs at 110 K at 0.1 MPa and at approx. 1 GPa at 298 K, corresponding to unit-cell volume of 865.0(7) and 824.6(18) Å³, respectively. “ does not make any sense, which is the underlying isochoric process? Also, it is unclear where the data is coming from.”*

Author reply: We are thankful for this comment. We rephrased the sentences to “The room-temperature ordering of the cage cations under pressure occurs when the unit-cell volume reaches 824.6(18) Å³. In turn, a fully ordered crystal structure of FMDMn at ambient pressure is observed at 110 K when the unit-cell volume is 865.0(7) Å³. This finding indicates that the ordering of the cage cations is not an isochoric process, and achieving an ordered structure through room-temperature compression requires a more significant reduction in unit-cell volume.”. The volume requirements for obtaining the ordered structure is taken from our CIF files. In turn, the ambient-pressure unit-cell volume characteristic for the ordered structure is taken from previous article on this material (doi.org/10.1021/ic500479e). We added the proper reference to this.

Comment i

Reviewer: *“The y-axis labelling of the insets in Figure 3a and b are wrong.”*

Author reply: All Figures were improved, as asked by the other Reviewers.

Comment j

Reviewer: *“I believe in deriving eq. 5 isothermal conditions are assumed, this should be added since eq. 4 is a function of p and T.”*

Author reply: We added this information to the text.

Comment k

Reviewer: *“P. 16 “V_a is complex temperature-independent material constant” – this is a generic sentence that arguably will be true for all HOIPs. At the same time, comparing the van-der-waal volume ratio with the ratio of V_a for both compounds underestimates the whole underlying complexity. The only scenario I see where both values would agree is if both A-cations would be spheres. A more thorough discussion on the interpretation of V_a based on chemical arguments should be given.”*

Author reply: The use of van-der-Waals volume does not require the cation to be of spherical shape. We agree with the Reviewer that one can use different descriptors, but they would all lead to the same conclusion. For example, Hirshfeld surface is commonly used in the analysis of crystal structures. One can easily calculate the volume under this surface. In general, the

volume increases with the increase in the size of the analyzed molecule/ion, which is actually an intuitive trend. The size of the Ace⁺ cation [NH₂-C(CH₃)-NH₂]⁺ is much smaller compared to, e.g., 1,4-diaminobutane [NH₂-CH₂-CH₂-CH₂-CH₂-NH₂]⁺ so we expect lower van der Waals volume or the volume under the Hirshfeld surface. Meanwhile, the activation volume of AceMn is significantly larger than that of DABZn. In the end, one arrives at exactly the same conclusion as that made on our van der Waals volume analysis- activation volume does not depend on the size of the reorienting cage cation. However, we agree with the Reviewer that a more detailed discussion on this matter would be needed. Therefore, we performed additional DFT calculations which shows that the activation volume is in fact a volumetric barrier for a relaxation process.

ANSWERS TO REVIEWER 2

General comment: *“The authors investigate the phase transition under different pressures. And they discussed the coupling motion in phase transition and revealed its effect in dielectric relaxation phenomenon. I must say sorry that these discussions are too trivial and some views are unclearly. Therefore, I can not recommend this paper for publication, in particular to Nature Commu.”*

Author response: We disagree with the Reviewer. The manuscript is divided into two parts: structural analysis of AceMn and FMDMn and dielectric studies on both compounds. Indeed, in the case of structural analysis, we are focused on the phase transitions in both compounds, as such studies were not done before. In this respect, we reveal a pressure-induced phase transition to a phase of higher symmetry which is unique in hybrid organic-inorganic perovskites (as marked also by Reviewer 1). Furthermore, we showed that the cage cations can experience a pressure-induced ordering without any phase transition, which constitute the mechanism not reported in the literature so far. Finally, our dielectric studies are not related to any phase transition analysis. We show that the relaxation times scale only with the activation volume and energy parameters. The volumetric changes during compression (and the related changes in hydrogen bond length and strength) have a minor influence. In fact, we show that both materials exhibit huge isothermal compressibility but, at the same time, the relaxation times shift to a small extent. This is not an obvious or trivial effect, as the relaxation process is commonly perceived to be highly dependent on the structure. In this manuscript we change the way of thinking, showing that it depends on the activation parameters, which are redefined here. The activation volume parameter is a volumetric barrier for a relaxation and, consequently, has little to do with the structure. Therefore, we show that one can tune the relaxation times (and thus dielectric constant) without changing the unit cell volume or other unit cell parameters by only varying the pressure-temperature conditions. We believe that this is not an obvious result.

Comment 1

Reviewer: *“The studies of coupling motion between A-site cations and inorganic framework have been investigated under low pressure (<1Gpa). The majority of the conclusions discussed by the author at high pressure are consistent with the phenomena observed in low-pressure phase transition (< 1GPa).”*

Author response: We agree with the Reviewer that the dielectric measurements were conducted mostly under low pressures. In the case of FMDMn, such measurements were, however, performed even up to 1.5 GPa (please refer to the inset in Figure 4a), where we could observe only residual traces of the relaxation process. The reason of almost total vanishing of the relaxation dynamics of A-site cations was explained by us by crystallography, specifically by the pressure evolution of the SOF parameter. As presented in Figure 1c, the crystal structure of FMDMn becomes basically structurally ordered above 1 GPa which is manifested by the SOF1 parameter approaching 1. The physical source of relaxation processes in dielectric spectra of such compounds is arising/vanishing orientation polarization, originating from collective motions of the A-site-cations. Consequently, with no reorienting A-site cations (SOF1=1), the relaxation process cannot emerge in dielectric spectra. As a result, the coupling motion between A-site cations and inorganic framework cannot be studied in FMDMn in the higher pressure regime. In the case of AceMn, a phase transition occurs around 1 GPa and the high-pressure structure is also ordered. As a result, studies of coupling motion between cage cations and inorganic framework cannot be done at higher pressures.

Comment 2

Reviewer: *“This effect, of negative linear compressibility (NLC), is a rare phenomenon among three-dimensional HOIPs”. In this paper, the author mentions the presence of the NLC phenomenon in the model systems, but there is a lack of thorough discussion regarding the relationship between this rare phenomenon at high pressure and the properties of material.”*

Author response: We appreciate your insightful observation regarding the Negative Linear Compressibility (NLC) phenomenon in the model systems discussed in our paper. We acknowledge that while we have mentioned the existence of NLC, there is indeed room for a more thorough discussion of the relationship between this crystal behaviour at high pressure. We believe that by providing this additional context and analysis, we can enhance the quality and impact of our paper. We appreciate your constructive critique, which will contribute to the refinement of our work.

Comment 3

Reviewer: *“Finally, our results provide compelling mathematical evidence that pressure-induced phase transitions in HOIPs are not only governed by the dynamics of movement of A-site cage cations, but it is a combined effect of ...” We think the control of pressure is an effective way to tune the volume of HOIPs. Further, the cage size and the cation reorientation have a strong coupling by H-bond. So that the volume can significantly control the orientation of A-site cations. We think this conclusion is trival and have been discussed in previous work. Maybe, in-depth theory are needed to revealed the coupling relationship.’*

Author reply: We are thankful for the reviewer for putting our attention on this matter. It is indeed tempting and natural to think that the relaxation dynamics in HOIPs should scale with the strength of hydrogen bonds or other interactions. As the volume decreases, these interactions increase their strength and the dynamics of reorienting cage cations slow down. This way of thinking is, however, deceptive and invalid. We wrote it in a more clear way in the corrected

manuscript. Namely, we plotted the relaxation times obtained during isothermal compression and isobaric cooling versus hydrogen bond length and unit cell volume. As there is no phase transition in the presented volumetric range, the unit cell volume scales with the the organic-inorganic cage size. Indeed, while both cooling and compressing, the hydrogen bond strength increases and the unit cell volume decreases, as expected. However, please note that there is a huge discrepancy in the relaxation times between these two thermodynamic paths.

In fact, the volumetric changes in compression (and related changes in hydrogen bond length and strength) have a minor influence on relaxation times. This is not an obvious or trivial effect, as the relaxation process is commonly perceived to be highly dependent on the structure. We explain it in more detail in the revised version of the manuscript based on the additional DFT calculations. In this manuscript we change the way of thinking, showing that it depends on the activation parameters, which are redefined here. The activation volume parameter is a volumetric barrier for a relaxation and, consequently, depends on more factors than the structural ones. Therefore, we show that one can tune the relaxation times (and thus dielectric constant) without changing the unit cell volume or other unit cell parameters by only varying the pressure-temperature conditions. We believe that this is not an obvious result.

Comment 4

Reviewer: “The pressure will most probably enhance the coupling of cation and framework. Some theoretical papers for coupling motion in HOIPs here: [Phys. Rev. B 94, 045201], [Phys. Rev. Lett. 122, 225701], [Phys. Rev. B 100, 09410].”

Author response: We are thankful to the Reviewer for providing us with the literature, which inspired us a lot. We carried out additional DFT calculations to understand better the mechanism of relaxation processes in our compounds, and to derive the definition of the activation volume parameter.

ANSWERS TO REVIEWER 3

Technical comments:

Reviewer: *“Overall, the manuscript needs an English fine polishing. This was actually hindering the view for the content and the review of the document. Overall, I had the impression that the intention was to produce well worded English, but the words were sought from the wrong sources and are actually not suitable for the intended meaning. I think many of the detailed comments later are simply due to this effect.”*

Author response: We are thankful for this remark. We improved the clarity, language and flow of the text.

Comment related to Abstract:

Reviewer: *“The abstract is written in report form, “we have done ...”. It should actually only contain the essentials of the major content of the paper and a scientific result summary.”*

Author response: We have rewritten the abstract so that it is focused on the essentials of the major content of the paper. Please note that we performed additional DFT calculations which shed new light on the relaxation dynamics of the cage cations and the definition of the activation volume parameter. As a result, the text of the introduction was changed not only grammatically, but it was also extended by new information.

Comment related to the Title:

Reviewer: *“I would omit the word “unveiling” in the title, it is meaningless towards the paper content. So I suggest: “Pressure-Driven Paradigm Shift: Activation Volume Effects in Hybrid Organic-Inorganic Compounds”*

Author response: We appreciate the Reviewer’s suggestion. We changed the title of the manuscript. However, concerning the additional comment of Reviewer 1, we had to rephrase it in a different way.

Comments on introduction:

Comment 1:

Reviewer: *“The authors are very familiar with the chemical formulae of the organic molecules, for the new reader at least the sum formula for both compounds must be given early on in the introduction and an image would always be helpful.”*

Author response: We are thankful for this remark. We added the following information to the introduction: *“In both compounds, the metal-organic framework is built of manganese Mn^{2+} cations and formate $HCOO^-$ anions, which form a 3D cage architecture. The cavities are occupied by A-site cage cations, i.e., formaamidinium (FMD^+ , $[NH_2-CH-NH_2]^+$) in the case of $FMDMn$, and acetamidinium (Ace^+ , $[NH_2-C(CH_3)-NH_2]^+$).”* Additionally, we provided the chemical formula of both compounds while listing the compounds for the first time.

Comment 2:

Reviewer: *“What do the authors mean by “phase over-compression”. This seems to make no sense here. I think it is an issue of English language.”*

Author response: We agree with the Reviewer that the wording was inappropriate. We removed it from the abstract and introduction completely.

Comment 3

Reviewer: *“There is discussion on “A-site cage ions or their components“ straight in the introduction without introducing the material that is actually concerned. This is confusing. We have 3 ions, so what is the A-site. Also these structures are more complex than simple perovskites, so site assignment is not trivial.”*

Author response: We added the basic information on the structure of three-dimensional hybrid metal-organic frameworks to the introduction, explaining step-by-step the porous architecture. The following sentences were added: “The key to their success is a unique tuneable cage-like host framework composed of metal cations coordinated by bridging ligands (organic or inorganic), with cavities occupied irreversibly or reversibly by guest cage molecules or ions. In this respect, particularly interesting is the use of HCOO⁻ anions as bridging ligands, as it takes place in hybrid formates.”. Please note, that we were additionally asked by Reviewer 1 to focus more on hybrid formates. Therefore, much of the introduction has been rewritten.

Comment 4

Reviewer: *“phenomenon” is the singular form, “phenomena” the plural, so write”*

Author response: We have carefully checked the introduction to avoid mistakes related to this words.

Comment 5

Reviewer: *“I personally do not like this sentence and similar statements when they do not apply: “Our results offer valuable insights for future HOIP research and the design of electronic and optoelectronic devices.” 3 GPa is a very high pressure and not available in any realistic device structure or industrial process. Thus, I would keep the argument along the gained scientific knowledge rather than technological impact of the findings. I do not deny that this material can be employed in technology, but not under high pressure”.*

Author response: We removed this sentence from the abstract.

Comment 6

Reviewer: *““plethora” is a very strong word and exaggerates the content of the paper.”*

Author reply: We changed this word to “numerous”.

Comments related to results

Comment 1

Reviewer: *“The temperature/pressure line in Table I is confusing”.*

Author response: Thank you for your valuable feedback. We understand that the temperature/pressure line in Table I was confusing. To address this, we have revised the table by separating the temperature and pressure into two distinct lines. This change is intended to facilitate readers in understanding the conditions under which the measurements were collected more clearly.

Comment 2

Reviewer: „*This sentence is strange: “righting the crystal direction [010] along which the H-bonds are aligned”*“

Author response: We apologize for the confusion. The correct term should indeed be "rigidity" instead of "righting." The revised sentence now reads:

"This provides additional support for the framework, indicating the rigidity of the crystal direction [010] along which the H-bonds are aligned, resulting in almost no compressibility along this direction up to 3.63 GPa (the parameter b is only reduced by about 1%)."

We appreciate your attention to detail and believe this correction enhances the clarity and accuracy of our manuscript.

Comment 3 nad 10

Reviewer: “*The authors state that there are MnO6-octahedra which rotate, where are these in this crystal structure or what do the authors mean?*”

Reviewer: „*Again MnO6-octahedra are mentioned. This is somehow very unclear. If Mn borrows two oxygen from each of the formate ions for the crystal structure, then this must be shown in a detailed image of the 3D structure. The crystal structure as shown in Figure 2 does not display how the formate ions order in the 3D structure. This was much better solved in the Physical Chemistry C-paper by the same authors.*”

Author reply: In the compounds we investigated, acetamidinium manganese(II) formate ($[\text{NH}_2\text{-C}(\text{CH}_3)\text{-NH}_2][\text{Mn}(\text{HCOO})_3]$, abbreviated as AceMn) and formamidinium manganese(II) formate ($[\text{NH}_2\text{-CH-NH}_2][\text{Mn}(\text{HCOO})_3]$, abbreviated as FMDMn), manganese (Mn) is octahedrally coordinated by oxygen atoms from the formate ions. Each Mn atom is surrounded by six oxygen atoms, with each formate ion contributing one oxygen atom to the coordination sphere of Mn. We understand that it is crucial to clearly convey the structural details. However, to avoid redundancy with previously published data, we have chosen not to include again schematic visualizations in the current manuscript. Instead, we provide a detailed textual explanation. We believe this approach, combined with the existing figures, sufficiently illustrates the coordination of the Mn atoms.

Comment 4

Reviewer: „*What is “its ‘wine-rack’-type strain”, this is not established terminology and also not self-explaining.*”

Author reply: This is a comment we cannot agree with the Reviewer. This term “‘wine-rack’-type strain” is well established in the literature and there are plenty of reports related to this effect.

Comment 5

Reviewer: *“The term “negative linear compressibility” is rather misleading. “Negative compressibility” under hydrostatic compression can only occur, if the crystal tilts. Thus, the seen effect is rather a shear effect and not negative compressibility which itself can only occur at a phase transition.”*

Author reply: Thank you for your comments regarding the term "negative linear compressibility" (NLC). We appreciate the opportunity to clarify this phenomenon.

One of the most basic principles of thermodynamics states that an object reduces its volume when the pressure increases ($\partial V/\partial p < 0$). In other words, the volume compressibility ($\beta_V = -1/V \times \partial V/\partial p|_{T=\text{const}}$) must be positive. This fundamental law applies to closed systems, i.e., objects that do not exchange their contents with their environment. In most crystalline materials, this volume reduction is achieved through a shortening in all three unit-cell dimensions. However, since individual unit-cell parameters are not thermodynamic variables, exceptions to this rule are possible. In some rare, anisotropic materials, one or two unit-cell dimensions increase under pressure. This behavior is called negative linear compressibility (NLC) or negative area compressibility (NAC).

NLC is a phenomenon where a material expands along one direction when subjected to hydrostatic pressure while still maintaining an overall volume reduction. This behavior occurs within a single phase and is due to unique structural arrangements that allow for anisotropic expansion. Such anisotropy is usually a consequence of a specific structural arrangement, and the most common structural feature associated with NLC is the wine-rack motif. This motif is present in our compound, formamidinium manganese(II) formate (FMDMn). In FMDMn, its hydrostatic compression leads to the elongation of one diagonal within the wine-rack, visible as the elongation of the unit-cell parameter, while others are reduced.

It is important to distinguish the NLC from changes in crystal size and shape during phase transitions. Phase transitions, such as those seen in ferroelastic materials, involve anomalous changes in the crystal structure and thus such effects can not be called NLC.

We hope this clarification resolves any misunderstanding regarding negative linear compressibility. We appreciate your feedback.

Comment 6

Reviewer: *„The authors use the term “void”. In crystallography this is defined as “Voids — small regions where there are no atoms, and which can be thought of as clusters of vacancies”. As far as I understand the sentence, this is not what the authors mean.”*

Author response: Indeed in crystallography, the term "void" typically refers to small regions within a crystal structure where there are no atoms. These voids, also known as interstitial spaces or clusters of vacancies, are an inherent part of all crystal structures. They play a crucial role in understanding the material's properties, such as its compressibility and stability.

Our use of the term "void" accurately reflects this definition. We refer to the spaces within the crystal structure that are not occupied by atoms. These voids are essential in explaining the structural properties and behavior of the materials at high-pressure. It is important to note that all crystalline materials contain voids, and recognizing their presence is critical for a comprehensive understanding of the materials behavior under compression.

Comment 7

Reviewer: „I think this statement is wrong: “In most cases, symmetry of molecular crystals is reduced in order to increase the number of degrees of freedom in structure, so it efficiently reduces the strain associated with the compression.” Which degrees of freedom are there supposed to be more? I think the term “degree of freedom” is not what the authors mean.”

Author response: Reducing symmetry in molecular crystals decreases the number of symmetry elements. A lower number of symmetry elements allows for greater flexibility in the structure, enabling it to accommodate strain more effectively under compression. This increased flexibility, or "degrees of freedom," allows the structure to adapt more easily to applied pressure, thereby reducing the strain.

When we refer to "degrees of freedom," we mean the ability of the structure to deform in response to external stress. In crystallography, this can involve various movements, such as translations, rotations, and distortions of the molecular components. A crystal with lower symmetry has fewer constraints on these movements, allowing it to better accommodate the strain induced by compression.

For example, in high-pressure environments, molecular crystals with lower symmetry can exhibit more complex and varied responses to pressure, such as changes in bond angles and lengths, without undergoing a phase transition. This ability to deform in a more versatile manner is what we refer to when discussing the increased degrees of freedom.

Comment 8

Reviewer: „Figure 2: the labels (d) and (e) are not mentioned in the figure caption.”

Author response: We changed all the Figures in the manuscript. Figure 2 has now panels from (a) to (f). The last three panels are put together in the caption as “(d-f) The structure of AceMn in phases I, II and III plotted along b axis.”

Comment 9

Reviewer: “What do the authors mean by “ambient-pressure H-bonding pattern“ ? Do you mean the special arrangement of the H-bonds? If so, please write so.”

Author response: By "ambient-pressure H-bonding pattern," we mean the specific arrangement of hydrogen bonds present in the structure of the material at room temperature and atmospheric pressure. This pattern is characteristic of the crystal structure under these conditions and is changed significantly under high pressure due to the effects of compression on the crystal structure.

Comment 11

Reviewer: “English: “progressing over a period of 2 weeks” should be after “reflection” in the same sentence”

Author reply: The sentence was corrected as suggested by the Reviewer.

Comment 12

Reviewer: “This sentence is in “Results” but to me it means a lot of interpretation: “The increased unit-cell dimension of residual phase III by ca. 1.3 % suggests a mosaic topology of amorphous highdensity regions.” It should be discussed in “Discussion”.”

Author reply: This is a comment we cannot agree with the Reviewer. The discussion section is devoted to the dielectrics and the motion of the cage cations in general. We are not focussing on the phase transition there and the highlighted sentence is somewhat related to this. We finish the results section with dielectric properties of the materials. As a result, the Reader could feel a little bit confused by making Him return to the basic structural behavior of the compounds.

Comment 13

Reviewer: “English: write: “This relaxation manifests itself as a bell-shaped ...”

Author reply: The wording has been changed as proposed by the Reviewer.

Comment 14

Reviewer: “How can you assume “under isochoric conditions” ? These materials undergo very high pressure loads, why should the volume not change? I do not understand this. You even gave compressibility coefficients.”

Author reply: In the studies, we analyze how the structures of AceMn and FMDMn changes with pressure at room temperature and temperature under ambient pressure. Ambient-pressure cooling and room-temperature compression both lead to reduction in the unit-cell volume. The lattice parameters change in the same way during both thermodynamic pathways. As a result, one can easily find such pairs of temperature and pressure under which the unit cell volume is the same. The Reviewer can see it on the right panel on the figure below. As marked by black arrows, there are such pairs pressure and temperature [(298K, pressure) vs (temperature, 0.1 MPa)] for the ambient-pressure cooling and room-temperature compression which are characterized by the same unit cell volume.

Comment 15

Reviewer: *“What do the authors mean by “due to contraction of the organic inorganic caverns”? Do they mean unoccupied space in the unit cell? This is unusual wording. A cavern is essentially a grotto. So I think this word is simply wrong.”*

Author reply: We changed the wording to “cage-like framework” as it is often called in the literature.

Comment 16

Reviewer: *“Replace “elongate” by “extend”. Elongate is a word that applies to space, not to time.”*

Author reply: We made the correction as indicated.

Comment 17

Reviewer: *“I do not understand the meaning of this sentence: “determines the p-T isochronous line of relaxation times”. What happens here at the same time? This sentence: “while $\tau_{\text{Max}(p)}$ remains constant under isochronous conditions“. That something stays constant in a single moment is trivially true. The following sentence is also not understandable: “that undergo isochronously, i.e., which are controlled by the dynamics of reorienting moieties “. As the word isochrones is used much in the manuscript and it is a very rare word and very specific, it must be clearly defined how it is used in the manuscript. One could e.g. use equation 6 to define such lines / times, but this must be done very clearly. This could be provided in the supplementary material, but it must then also be referenced in the main manuscript. I do not understand this conclusion: “However, provided that the dielectric relaxation phenomena in such HOIPs are connected to the electric-field induced reorientation of the organic A-site cage cations, it can be concluded that the observed pressure-induced phase transitions in these materials (e.g., AceMn) cannot be governed solely by the dynamics of cage cations.” This is now a scientific matter, not English. The dielectric response contains all contributions, namely cage motion, molecular rotation and relative shift of the positively and negatively charged sub-lattices. It is only a question in which frequency ranges these respective mechanisms become dominant in the dielectric response.”*

Author reply: Our intention was to show that relaxation times cannot be the sole determinant governing the conditions for a phase transition. However, we agree with the Reviewer and also with Reviewer 1, that discussion on this matter is too complicated, not straightforward and can be considered as misleading. That is why we completely removed it from the manuscript. Most of the contribution to the relaxation process comes from the cage cations in hybrid perovskites of centrosymmetric symmetry. This is because the formate ions are quite rigid and only slightly reorient when the cage cation rotate (see the performed DFT calculations). As a result their dipole moments from the formate ions almost cancel out. Most likely their motion will be visible at much higher frequencies (terahertz regime) as these are small-angular motions.

Comment 18

Reviewer: *“There are no red points in Fig. 4 (c). There is black and orange”*

Author reply: All the figures were reorganized.

Comment 19

Reviewer: ““According to the calculations made for the reference τ_{Max} equal to $1 \mu s$ based on the equations of state, the elevation of pressure up to 300 MPa shifts the relaxation peak as much as a decrease in temperature by about 4.5 K for FMDMn and 11 K for AceMn (see Fig. S2).” This statement is rather surprising. I urge the authors to reproduce the data / check the calculations to make sure that this statement is correct.”

Author reply: We agree with the Reviewer that this is indeed a surprising result. The typical way of thinking is that if the crystal lattice is soft and contracts much under pressure, the relaxation times should also shift much. However, counterintuitively, the strength of hydrogen bonds and other interactions is not the sole determinant governing the relaxation dynamics. In fact, motion of the cage cations require the formate ions to reorient slightly as indicated by DFT calculations. This, in turn, is regulated by vibrations (phonons) and thermal energy predominantly as these are only small-angular motions. As a result, at room temperature, the formate ions can reorient even under high pressure as there is enough thermal energy delivered to the system. Consequently, changes in the relaxation times are small. Besides, we show that the relaxation process is governed by two barriers: energy and volumetric. The second is small as only small-angular rearrangements are required to allow the FMD^+ and Ace^+ cations reorienting. In this case, the pressure-induced changes at constant temperature have to be small following the equation of state. This effect has been nicely documented previously for 1,4-diaminobutane zinc formate (DABZn) in 10.1039/D0TC04047A (see figure below). Therefore, we added the following sentence to the manuscript: “*In the case of 1,4-diaminobutane zinc formate (DABZn), isothermal compression up to 1.7 GPa at 302 K exerts a similar effect on relaxation times as a mere 17 K decrease in temperature (to 285 K).*”

Our results are very comparable to DABZn, as published before. Therefore, with the unobvious results, this manuscript aims at changing the way of thinking about relaxation processes in hybrid perovskites. We believe, that due to that fact the Reviewer would find it suitable for Nature Communications.

Comment 20

Reviewer: ““ V_a is a complex temperature-independent material constant. “ What is so complex about a constant? If something is constant to me it appears more simple than something that changes.”

Author reply: It is hard not to agree with the Reviewer. We have rewritten completely the discussion and conclusions, avoiding the word “complex”.

Comment 21

Reviewer: “The summary is too long (two pages). It should be reduced to the essentials that are found in the paper and not summarize everything. A conclusion is more appropriate here than a summary. “In summary, ambient- and high-pressure X-ray diffraction and broadband dielectric studies were performed on model formamidinium manganese(II) formate (FMDMn) and acetamidinium manganese(II) formate (AceMn) to unveil their exceptional features and derive fundamental principles governing the behavior of these prototypical HOIPs under various pressure regimes. “ This sentence is completely meaningless in a summary. The reader does not want to read how important the work is, but wants to understand from the data what is significant. The summary should be fully content based and not blabla.”

Author reply: We have rewritten the summary section, shortening and improving it.

Comment 22

Reviewer: ““Moreover, V_a is a complex temperature-independent material constant for AceMn and FMDMn. It not only governs the volume requirements for the movement of the cage A-site cations in response to external stimuli but also incorporates information about the stiffness of the cavities.” How does it “incorporate” information about stiffness? This is not explained in the manuscript.”

Author reply: We carried out additional DFT calculations to derive a more precise definition of V_a . Based on them, we showed that V_a should be treated as a volumetric barrier for the relaxation of cage cations. Motion of the cage cations require the metal-formate framework to reshape. Consequently, V_a depends much on the stiffness of this skeleton. With increasing stiffness of the metal-formate framework, the cage cations are less affected by the external pressure as the H-bond length changes less. The volumetric contribution to the relaxation times is reduced in this way and according to the pressure form of the Arrhenius transition, V_a value decreases. We discussed it in detail in the revised version of our manuscript and illustrated this effect schematically in Figure 6d.

Comment 23

Reviewer: ““Finally, our results provide compelling mathematical evidence that pressure-induced phase transitions in HOIPs are not only governed by the dynamics of movement of A-site cage cations, but it is a combined effect of appearing...”. I do not see any mathematical evidence in this manuscript. It is all experimental with a few fit functions. There are no mathematical derivations, it is all thermodynamics.”

Author response: As we wrote before, we rephrased the whole conclusion section. We removed this sentence.

Comment 24

Reviewer: *““This finding opens up new avenues for research into the dynamic behaviour of A-site cage cations under pressure, urging further exploration of the underlying physics in these materials. Consequently, our study unravels features of AceMn and FMDMn and establishes fundamental principles governing the behaviour of formate-based HOIPs under varying pressure regimes.” Two completely meaningless sentences. Sentences like this one must be removed they only unnecessarily extend the paper and thus make it unreadable. Only concise clear texts are pleasant to read.”*

Author response: We appreciate this comment. We removed this text from the manuscript.

ANSWERS TO REVIEWER 4

General comment

Reviewer: *“The manuscript describes a thorough reexamination on the effect of pressure on the structural behavior of two HOIPS – formadinium manganese formate, and acetamidinium manganese formate. Fundamentally, I believe this work presents careful data analysis that is of interest to the broad readership of Nature Communications. I believe their interpretation of the single crystal/dielectric data is good and provides a useful metric for examining these intriguing compounds. However, there are areas of the manuscript that need addressing. Issues are predominantly the presentation of data, but there is also a potentially glaring gloss over of Raman data mentioned. Without that data included, though the SC/dielectric data presented is the main focus of the work, the manuscript likely needs another round of examination. Figure S4 of the Raman data is mentioned, but not included in the manuscript. I am uncertain if the authors are referencing Raman experiments formerly published that they are including in this manuscript, or if there are new Raman experiments that confirm or align with that previously published data / the authors own crystallographic/dielectric observations. I believe if the authors are referencing and analytically leaning on the Raman data, that this data needs to not only be included in the SI, but be more thoroughly presented/analyzed and referenced. I believe that the authors own concluding paragraph, which suggests the impacts of “vibrations” may be rooted in this spectroscopy, which is currently only nodded to.”*

Author response: We are thankful for this remark. We added the high-pressure Raman data to the manuscript and discussed them. The data fully support our structural analysis, as presented in the revised version of the manuscript.

Comment 1

Reviewer: *“Figure 3 needs to be reworked as currently the insets are quite difficult to read. For example, in Figure 3a, the caption refers to the open stars for calculated pressure dependence, but without excessive zooming in, the symbols are very hard to see relative to the presented data. This could be a combination of the quality of PDF provided to the reviewers, but I believe may also speak to potential what the final reader may encounter. Along those lines, Figure 4 also needs addressing, as it is unclear which way data was taken in panels a and b. I believe from the text that we’re moving right to left, but this should be inherent in the figure. In general, I would suggest that the reviewers go back and look at all figures and evaluate if data is presented in a way that reflects what it should to the reader without digging.”*

Author response: We improved all the Figures included in the manuscript. Please note that we added some new analyses as requested by other Reviewers. As a result, the whole manuscript and the Figures have been reorganized.”

Comment 2

Reviewer: *“I thank the reviewers for their incredible transparency in the amount of .cif's deposited in the CCDC. However, given that the authors are depositing 10's of single crystal data sets, there should be an accompanying table in the SI relating what specific ccdc number*

is what specific .cif (experimental condition). This is only an issue given the sheer number of .cifs (because if there were say just 3, they would be trivial to sort through). Doing so will give the eventual readers of this manuscript the capacity to examine this data without undue burden.”

Author response: We sincerely appreciate your valuable feedback and your acknowledgment of the transparency in depositing .cifs in the CCDC. We understand your concern, given the significant number of .cifs being deposited. To address this, we have added tables in the Supplementary Information (SI) to provide a clear mapping between specific CCDC numbers and their corresponding .cif files, including the experimental conditions. The tables have been included in the SI with labels Table S1-S8, and they associate the CIF numbers 2287917-2287940 with ACEMn and 2287942-2287953 with FMDMn. These tables aim to facilitate easy access and examination of the crystallographical data, ensuring that readers can navigate and utilize this information efficiently without undue burden.

Comment 3

Reviewer: *“There are minor typos throughout – for example, I believe the data presented in Table 1 (ratios of the Z phases), as well as minor grammatical issues scattered through the manuscript. This stems back to my comments on the figures, and the authors should take the time to polish their work and present it in a way that is commensurate with their data analysis.”*

Author response: We are thankful for bringing our attention to this matter. We carefully checked the entire revised manuscript to avoid any typos.

REVIEWERS' COMMENTS

Reviewer #2 (Remarks to the Author):

The reviewers have properly addressed my previous concerns and I therefore recommend it for publications.

Reviewer #3 (Remarks to the Author):

I still do not like the title.

The authors should seek for a neutral and conclusive title.

As Nature Communications attracts enough readership just by the journal, it is not necessary to try to embellish things through the title.

Otherwise, the manuscript has improved much. Even though I did not ask for this, but I think the DFG support is rather helpful for the manuscript.